# LMC: Fast Training of GNNs via subgraph-wise sampling with Provable Convergence

**Zhihao Shi** [1] , **Xize Liang** [1], **Jie Wang**[*1],
[1] University of Science and Technology of China

## Abstract

The message passing-based graph neural networks (GNNs) have achieved great success in many real-world applications. However, training GNNs on large-scale graphs suffers from the well-known *neighbor explosion* problem, i.e., the exponentially increasing dependencies of nodes with the number of message passing layers. Subgraph-wise sampling methods—a promising class of mini-batch training techniques—discard messages outside the mini-batches in backward passes to avoid the neighbor explosion problem at the expense of gradient estimation accuracy. This poses significant challenges to their convergence analysis and convergence speeds, which seriously limits their reliable real-world applications. To address this challenge, we propose a novel subgraph-wise sampling method with a convergence guarantee, namely **L**ocal **M**essage **C**ompensation (LMC). To the best of our knowledge, LMC is the *first* subgraph-wise sampling method with provable convergence. The key idea of LMC is to retrieve the discarded messages in backward passes based on a message passing formulation of backward passes. By efficient and effective compensations for the discarded messages in both forward and backward passes, LMC computes accurate mini-batch gradients and thus accelerates convergence. We further show that LMC converges to first-order stationary points of GNNs. Experiments on large-scale benchmark tasks demonstrate that LMC significantly outperforms state-of-the-art subgraph-wise sampling methods in terms of efficiency.

## 1 Introduction

Graph neural networks (GNNs) are powerful frameworks that generate node embeddings for graphs via the iterative message passing (MP) scheme (Hamilton, 2020). At each MP layer, GNNs aggregate messages from each node's neighborhood and then update node embeddings based on aggregation results. Such a scheme has achieved great success in many real-world applications involving graph-structured data, such as search engines (Brin & Page, 1998), recommendation systems (Fan et al., 2019), materials engineering (Gostick et al., 2016), molecular property prediction (Moloi & Ali, 2005; Kearnes et al., 2016), and combinatorial optimization (Wang et al., 2023).

However, the iterative MP scheme poses challenges to training GNNs on large-scale graphs. One commonly-seen approach to scale deep models to arbitrarily large-scale data with limited GPU memory is to approximate full-batch gradients by mini-batch gradients. Nevertheless, for the graph-structured data, the computational costs for computing the loss across a mini-batch of nodes and the corresponding mini-batch gradients are expensive due to the well-known *neighbor explosion* problem. Specifically, the embedding of a node at the $k$-th MP layer recursively depends on the embeddings of its neighbors at the $(k-1)$-th MP layer. Thus, the complexity grows exponentially with the number of MP layers.

To deal with the neighbor explosion problem, recent works propose various sampling techniques to reduce the number of nodes involved in message passing (Ma & Tang, 2021). For example, node-wise (Hamilton et al., 2017; Chen et al., 2018a) and layer-wise (Chen et al., 2018b; Zou et al., 2019; Huang et al., 2018) sampling methods recursively sample neighbors over MP layers to estimate node embeddings and corresponding mini-batch gradients. Unlike the recursive fashion, subgraph-wise sampling methods (Chiang et al., 2019; Zeng et al., 2020; Fey et al., 2021; Zeng et al., 2021) adopt a

---

*Corresponding author: jiewangx@ustc.edu.cn

cheap and simple one-shot sampling fashion, i.e., sampling the same subgraph constructed based on a mini-batch for different MP layers. By discarding messages outside the mini-batches, subgraph-wise sampling methods restrict message passing to the mini-batches such that the complexity grows linearly with the number of MP layers. Moreover, subgraph-wise sampling methods are applicable to a wide range of GNN architectures by directly running GNNs on the subgraphs constructed by the sampled mini-batches (Fey et al., 2021). Because of these advantages, subgraph-wise sampling methods have recently drawn increasing attention.

Despite the empirical success of subgraph-wise sampling methods, discarding messages outside the mini-batch sacrifices the gradient estimation accuracy, which poses significant challenges to their convergence behaviors. First, recent works (Chen et al., 2018a; Cong et al., 2020) demonstrate that the inaccurate mini-batch gradients seriously hurt the convergence speeds of GNNs. Second, in Section 7.3, we demonstrate that many subgraph-wise sampling methods are difficult to resemble full-batch performance under small batch sizes, which we usually use to avoid running out of GPU memory in practice. These issues seriously limit the real-world applications of GNNs.

In this paper, we propose a novel subgraph-wise sampling method with a convergence guarantee, namely **L**ocal **M**essage **C**ompensation (LMC), which uses efficient and effective compensations to correct the biases of mini-batch gradients and thus accelerates convergence. To the best of our knowledge, LMC is the *first* subgraph-wise sampling method with provable convergence. Specifically, we first propose unbiased mini-batch gradients for the one-shot sampling fashion, which helps decompose the gradient computation errors into two components: the bias from the discarded messages and the variance of the unbiased mini-batch gradients. Second, based on a message passing formulation of backward passes, we retrieve the messages discarded by existing subgraph-wise sampling methods during the approximation to the unbiased mini-batch gradients. Finally, we propose efficient and effective compensations for the discarded messages with a combination of incomplete up-to-date messages and messages generated from historical information in previous iterations, avoiding the exponentially growing time and memory consumption. An appealing feature of the resulting mechanism is that it can effectively correct the biases of mini-batch gradients, leading to accurate gradient estimation and the speed-up of convergence. We further show that LMC converges to first-order stationary points of GNNs. Notably, the convergence of LMC is based on the interactions between mini-batch nodes and their 1-hop neighbors, without the recursive expansion of neighborhoods to aggregate information far away from the mini-batches. Experiments on large-scale benchmark tasks demonstrate that LMC significantly outperforms state-of-the-art subgraph-wise sampling methods in terms of efficiency. Moreover, under small batch sizes, LMC outperforms the baselines and resembles the prediction performance of full-batch methods.

## 2 RELATED WORK

In this section, we discuss some works related to our proposed method.

**Subgraph-wise Sampling Methods.** Subgraph-wise sampling methods sample a mini-batch and then construct the same subgraph based on it for different MP layers (Ma & Tang, 2021). For example, Cluster-GCN (Chiang et al., 2019) and GraphSAINT (Zeng et al., 2020) construct the subgraph induced by a sampled mini-batch. They encourage connections between the sampled nodes by graph clustering methods (e.g., METIS (Karypis & Kumar, 1998) and Graclus (Dhillon et al., 2007)), edge, node, or random-walk-based samplers. GNNAutoScale (GAS) (Fey et al., 2021) and MVS-GNN (Cong et al., 2020) use historical embeddings to generate messages outside a sampled subgraph, maintaining the expressiveness of the original GNNs.

**Recursive Graph Sampling Methods.** Both node-wise and layer-wise sampling methods recursively sample neighbors over MP layers and then construct different computation graphs for each MP layer. Node-wise sampling methods (Hamilton et al., 2017; Chen et al., 2018a) aggregate messages from a small subset of sampled neighborhoods at each MP layer to decrease the bases in the exponentially increasing dependencies. To avoid the exponentially growing computation, layer-wise sampling methods (Chen et al., 2018b; Zou et al., 2019; Huang et al., 2018) independently sample nodes for each MP layer and then use importance sampling to reduce variance, resulting in a constant sample size in each MP layer.

**Pre-Processing Methods.** Another line for scalable graph neural networks is to develop pre-processing Methods. They aggregate the raw input features and then take the pre-processing features as input into subsequent models. As the aggregation has no parameters, they can use stochastic gradient descent to train the subsequent models without the neighbor explosion problem. While

they are efficient in training and inference, they are not applicable to powerful GNNs with a trainable aggregation process.

**Historical Values as an Affordable Approximation.** The historical values are affordable approximations of the exact values in practice. However, they suffer from frequent data transfers to/from the GPU and the staleness problem. For example, in node-wise sampling, VR-GCN (Chen et al., 2018a) uses historical embeddings to reduce the variance from neighbor sampling (Hamilton et al., 2017). GAS (Fey et al., 2021) proposes a concurrent mini-batch execution to transfer the active historical embeddings to and from the GPU, leading to comparable runtime with the standard full-batch approach. GraphFM-IB and GraphFM-OB (Yu et al., 2022) apply a momentum step on historical embeddings for node-wise and subgraph-wise sampling methods with historical embeddings, respectively, to alleviate the staleness problem. Both LMC and GraphFM-OB use the node embeddings in the mini-batch to alleviate the staleness problem of the node embeddings outside the mini-batch. We discuss the main differences between LMC and GraphFM-OB in Appendix C.1.

## 3 PRELIMINARIES

We introduce notations and graph neural networks in Sections 3.1 and 3.2, respectively.

### 3.1 NOTATIONS

A graph $\mathcal{G} = (\mathcal{V}, \mathcal{E})$ is defined by a set of nodes $\mathcal{V} = \{v_1, v_2, \ldots, v_n\}$ and a set of edges $\mathcal{E}$ among these nodes. The set of nodes consists of labeled nodes $\mathcal{V}_L$ and unlabeled nodes $\mathcal{V}_U := \mathcal{V} \setminus \mathcal{V}_L$. Let $(v_i, v_j) \in \mathcal{E}$ denote an edge going from node $v_i \in \mathcal{V}$ to node $v_j \in \mathcal{V}$, $\mathcal{N}(v_i) = \{v_j \in \mathcal{V} | (v_i, v_j) \in \mathcal{E}\}$ denote the neighborhood of node $v_i$, and $\overline{\mathcal{N}}(v_i)$ denote $\mathcal{N}(v_i) \cup \{v_i\}$. We assume that $\mathcal{G}$ is undirected, i.e., $v_j \in \mathcal{N}(v_i) \Leftrightarrow v_i \in \mathcal{N}(v_j)$. Let $\mathcal{N}(\mathcal{S}) = \{v \in \mathcal{V} | (v_i, v_j) \in \mathcal{E}, v_i \in \mathcal{S}\}$ denote the neighborhoods of a set of nodes $\mathcal{S}$ and $\overline{\mathcal{N}}(\mathcal{S})$ denote $\mathcal{N}(\mathcal{S}) \cup \mathcal{S}$. For a positive integer $L$, $[L]$ denotes $\{1, \ldots, L\}$. Let the boldface character $\mathbf{x}_i \in \mathbb{R}^{d_x}$ denote the feature of node $v_i$ with dimension $d_x$. Let $\mathbf{h}_i \in \mathbb{R}^d$ be the $d$-dimensional embedding of the node $v_i$. Let $\mathbf{X} = (\mathbf{x}_1, \mathbf{x}_2, \ldots, \mathbf{x}_n) \in \mathbb{R}^{d_x \times n}$ and $\mathbf{H} = (\mathbf{h}_1, \mathbf{h}_2, \ldots, \mathbf{h}_n) \in \mathbb{R}^{d \times n}$. We also denote the embeddings of a set of nodes $\mathcal{S} = \{v_{i_k}\}_{k=1}^{|\mathcal{S}|}$ by $\mathbf{H}_{\mathcal{S}} = (\mathbf{h}_{i_k})_{k=1}^{|\mathcal{S}|} \in \mathbb{R}^{d \times |\mathcal{S}|}$. For a $p \times q$ matrix $\mathbf{A} \in \mathbb{R}^{p \times q}$, $\vec{\mathbf{A}} \in \mathbb{R}^{pq}$ denotes the vectorization of $\mathbf{A}$, i.e., $\mathbf{A}_{ij} = \vec{\mathbf{A}}_{i+(j-1)p}$. We denote the $j$-th columns of $\mathbf{A}$ by $\mathbf{A}_j$.

### 3.2 GRAPH NEURAL NETWORKS

For the semi-supervised node-level prediction, Graph Neural Networks (GNNs) aim to learn node embeddings $\mathbf{H}$ with parameters $\Theta$ by minimizing the objective function $\mathcal{L} = \frac{1}{|\mathcal{V}_L|} \sum_{i \in \mathcal{V}_L} \ell_w(\mathbf{h}_i, y_i)$ such that $\mathbf{H} = \mathcal{GNN}(\mathbf{X}, \mathcal{E}; \Theta)$, where $\ell_w$ is the composition of an output layer with parameters $w$ and a loss function.

GNNs follow the message passing framework in which vector messages are exchanged between nodes and updated using neural networks. An $L$-layer GNN performs $L$ message passing iterations with different parameters $\Theta = (\theta^l)_{l=1}^L$ to generate the final node embeddings $\mathbf{H} = \mathbf{H}^L$ as

$$\mathbf{H}^l = f_{\theta^l}(\mathbf{H}^{l-1}; \mathbf{X}), \ l \in [L], \tag{1}$$

where $\mathbf{H}^0 = \mathbf{X}$ and $f_{\theta^l}$ is the message passing function of the $l$-th layer with parameters $\theta^l$.

The message passing function $f_{\theta^l}$ follows an *aggregation* and *update* scheme, i.e.,

$$\mathbf{h}_i^l = u_{\theta^l}\left(\mathbf{h}_i^{l-1}, \mathbf{m}_{\mathcal{N}(v_i)}^{l-1}, \mathbf{x}_i\right); \quad \mathbf{m}_{\mathcal{N}(v_i)}^{l-1} = \oplus_{\theta^l}\left(\left\{g_{\theta^l}(\mathbf{h}_j^{l-1}) \mid v_j \in \mathcal{N}(v_i)\right\}\right), \ l \in [L], \tag{2}$$

where $g_{\theta^l}$ is the function generating *individual messages* for each neighbor of $v_i$ in the $l$-th message passing iteration, $\oplus_{\theta^l}$ is the aggregation function mapping a set of messages to the final message $\mathbf{m}_{\mathcal{N}(v_i)}^{l-1}$, and $u_{\theta^l}$ is the update function that combines previous node embedding $\mathbf{h}_i^{l-1}$, message $\mathbf{m}_{\mathcal{N}(v_i)}^{l-1}$, and features $\mathbf{x}_i$ to update node embeddings.

## 4 MESSAGE PASSING IN BACKWARD PASSES

In Section 4.1, we introduce the gradients of GNNs and formulate the backward passes as message passing. Then we propose backward SGD, which is an SGD variant, in Section 4.2.

### 4.1 BACKWARD PASSES AND MESSAGE PASSING FORMULATION

The gradient $\nabla_w \mathcal{L}$ is easy to compute and we hence introduce the chain rule to compute $\nabla_\Theta L$ in this section, where $\Theta = (\theta^l)_{l=1}^L$. Let $\mathbf{V}^l \triangleq \nabla_{\mathbf{H}^l} \mathcal{L}$ for $l \in [L]$ be auxiliary variables. It is easy to compute

$\vec{\mathbf{V}}^L = \nabla_{\vec{\mathbf{H}}^L} \mathcal{L} = \nabla_{\vec{\mathbf{H}}} \mathcal{L}$. By the chain rule, we iteratively compute $\mathbf{V}^l$ based on $\mathbf{V}^{l+1}$ as

$$\vec{\mathbf{V}}^l = \vec{\phi}_{\theta^{l+1}}(\mathbf{V}^{l+1}) \triangleq (\nabla_{\vec{\mathbf{H}}^l} \vec{f}_{\theta^{l+1}}) \vec{\mathbf{V}}^{l+1} \tag{3}$$

and

$$\mathbf{V}^l = \phi_{\theta^{l+1}} \circ \cdots \circ \phi_{\theta^L}(\mathbf{V}^L). \tag{4}$$

Then, we compute the gradient $\nabla_{\theta^l} \mathcal{L} = (\nabla_{\theta^l} \vec{f}_{\theta^l}) \vec{\mathbf{V}}^l$, $l \in [L]$ by using autograd packages for vector-Jacobian product.

We formulate backward passes, i.e., the processes of iterating Equation (3), as message passing. To see this, we need to notice that Equation (3) is equivalent to

$$\mathbf{V}_i^l = \sum_{v_j \in \mathcal{N}(v_i)} \left( \nabla_{\mathbf{h}_i^l} u_{\theta^{l+1}}(\mathbf{h}_j^l, \mathbf{m}_{\mathcal{N}(v_j)}^l, \mathbf{x}_j) \right) \mathbf{V}_j^{l+1}, \ i \in [n], \tag{5}$$

where $\mathbf{V}_k^l$ is the $k$-th column of $\mathbf{V}^l$ and $\mathbf{m}_{\mathcal{N}(v_j)}^l$ is a function of $\mathbf{h}_i^l$ defined in Equation (2). Equation (5) uses $\left( \nabla_{\mathbf{h}_i^l} u_{\theta^{l+1}}(\mathbf{h}_j^l, \mathbf{m}_{\mathcal{N}(v_j)}^l, \mathbf{x}_j) \right) \mathbf{V}_j^{l+1}$, sum aggregation, and the identity mapping as the generation function, the aggregation function, and the update function, respectively.

## 4.2 BACKWARD SGD

In this section, we develop an SGD variant—backward SGD, which provides unbiased gradient estimations based on the message passing formulation of backward passes. Backward SGD is the basis of our proposed subgraph-wise sampling method, i.e., LMC, in Section 5.

Given a sampled mini-batch $\mathcal{V}_\mathcal{B}$, suppose that we have computed exact node embeddings $(\mathbf{H}_{\mathcal{V}_\mathcal{B}}^l)_{l=1}^L$ and auxiliary variables $(\mathbf{V}_{\mathcal{V}_\mathcal{B}}^l)_{l=1}^L$ of nodes in $\mathcal{V}_\mathcal{B}$. To simplify the analysis, we assume that $\mathcal{V}_\mathcal{B}$ is uniformly sampled from $\mathcal{V}$ and the corresponding set of labeled nodes $\mathcal{V}_{L_\mathcal{B}} := \mathcal{V}_\mathcal{B} \cap \mathcal{V}_L$ is uniformly sampled from $\mathcal{V}_L$. When the sampling is not uniform, we use the normalization technique (Zeng et al., 2020) to enforce the assumption (please see Appendix A.3.1).

First, backward SGD computes the mini-batch gradient $\mathbf{g}_w(\mathcal{V}_\mathcal{B})$ for parameters $w$ by the derivative of mini-batch loss $\mathcal{L}_{\mathcal{V}_\mathcal{B}} = \frac{1}{|\mathcal{V}_{L_\mathcal{B}}|} \sum_{v_j \in \mathcal{V}_{L_\mathcal{B}}} \ell_w(\mathbf{h}_j, y_j)$ as

$$\mathbf{g}_w(\mathcal{V}_\mathcal{B}) = \frac{1}{|\mathcal{V}_{L_\mathcal{B}}|} \sum_{v_j \in \mathcal{V}_{L_\mathcal{B}}} \nabla_w \ell_w(\mathbf{h}_j, y_j). \tag{6}$$

Then, backward SGD computes the mini-batch gradient $\mathbf{g}_{\theta^l}(\mathcal{V}_\mathcal{B})$ for parameters $\theta^l$ as

$$\mathbf{g}_{\theta^l}(\mathcal{V}_\mathcal{B}) = \frac{|\mathcal{V}|}{|\mathcal{V}_\mathcal{B}|} \sum_{v_j \in \mathcal{V}_\mathcal{B}} \left( \nabla_{\theta^l} u_{\theta^l}(\mathbf{h}_j^{l-1}, \mathbf{m}_{\mathcal{N}(v_j)}^{l-1}, \mathbf{x}_j) \right) \mathbf{V}_j^l, \ l \in [L]. \tag{7}$$

Note that the mini-batch gradients $\mathbf{g}_{\theta^l}(\mathcal{V}_\mathcal{B})$ for different $l \in [L]$ are based on the same mini-batch $\mathcal{V}_\mathcal{B}$, which facilitates designing subgraph-wise sampling methods based on backward SGD. Another appealing feature of backward SGD is that the mini-batch gradients $\mathbf{g}_w(\mathcal{V}_\mathcal{B})$ and $\mathbf{g}_{\theta^l}(\mathcal{V}_\mathcal{B})$, $l \in [L]$ are unbiased, as shown in the following theorem. Please see Appendix D.1 for the detailed proof.

**Theorem 1.** *Suppose that a mini-batch $\mathcal{V}_\mathcal{B}$ is uniformly sampled from $\mathcal{V}$ and the corresponding labeled nodes $\mathcal{V}_{L_\mathcal{B}} = \mathcal{V}_\mathcal{B} \cap \mathcal{V}_L$ is uniformly sampled from $\mathcal{V}_L$. Then the mini-batch gradients $\mathbf{g}_w(\mathcal{V}_\mathcal{B})$ and $\mathbf{g}_{\theta^l}(\mathcal{V}_\mathcal{B})$, $l \in [L]$ in Equations (6) and (7) are unbiased.*

## 5 LOCAL MESSAGE COMPENSATION

The exact mini-batch gradients $\mathbf{g}_w(\mathcal{V}_\mathcal{B})$ and $\mathbf{g}_{\theta^l}(\mathcal{V}_\mathcal{B})$, $l \in [L]$ computed by backward SGD depend on exact embeddings and auxiliary variables of nodes in the mini-batch $\mathcal{V}_\mathcal{B}$ rather than the whole graph. However, backward SGD is not scalable, as the exact $(\mathbf{H}_{\mathcal{V}_\mathcal{B}}^l)_{l=1}^L$ and $(\mathbf{V}_{\mathcal{V}_\mathcal{B}}^l)_{l=1}^L$ are expensive to compute due to the *neighbor explosion* problem.

In this section, to deal with the neighbor explosion problem, we develop a novel and scalable subgraph-wise sampling method for GNNs, namely **L**ocal **M**essage **C**ompensation (LMC). LMC first efficiently estimates $(\mathbf{H}_{\mathcal{V}_\mathcal{B}}^l)_{l=1}^L$ and $(\mathbf{V}_{\mathcal{V}_\mathcal{B}}^l)_{l=1}^L$ by convex combinations of the *incomplete up-to-date values* and the *historical values*, and then computes the mini-batch gradients as shown in

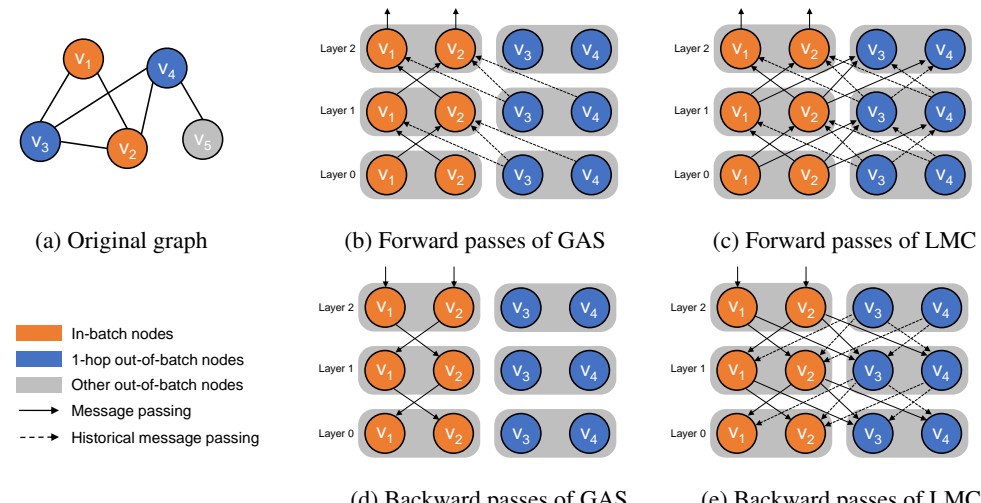

(a) Original graph  (b) Forward passes of GAS  (c) Forward passes of LMC

(d) Backward passes of GAS  (e) Backward passes of LMC

Figure 1: Comparison of LMC with GNNAutoScale (GAS) (Fey et al., 2021). (a) shows the original graph with in-batch nodes, 1-hop out-of-batch nodes, and other out-of-batch nodes in orange, blue, and grey, respectively. (b) and (d) show the computation graphs of forward passes and backward passes of GAS, respectively. (c) and (e) show the computation graphs of forward passes and backward passes of LMC, respectively.

Equations (6) and (7). We show that LMC converges to first-order stationary points of GNNs in Section 6. In Algorithm 1 and Section 6, we denote a value in the $l$-th layer at the $k$-th iteration by $(\cdot)^{l,k}$, but elsewhere we omit the superscript $k$ and denote it by $(\cdot)^l$.

In each training iteration, we sample a mini-batch of nodes $\mathcal{V}_\mathcal{B}$ and propose to approximate values, i.e., node embeddings and auxiliary variables, outside $\mathcal{V}_\mathcal{B}$ by convex combinations of *historical values*, denoted $\overline{\mathbf{H}}^l_{\mathcal{V}\setminus\mathcal{V}_\mathcal{B}}$ and $\overline{\mathbf{V}}^l_{\mathcal{V}\setminus\mathcal{V}_\mathcal{B}}$, and *incomplete up-to-date values*, denoted $\widetilde{\mathbf{H}}^l_{\mathcal{V}\setminus\mathcal{V}_\mathcal{B}}$ and $\widetilde{\mathbf{V}}^l_{\mathcal{V}\setminus\mathcal{V}_\mathcal{B}}$.

In forward passes, we initialize the *temporary embeddings* for $l = 0$ as $\widehat{\mathbf{H}}^0 = \mathbf{X}$ and update historical embeddings of nodes in $\mathcal{V}_\mathcal{B}$, i.e., $\overline{\mathbf{H}}^l_{\mathcal{V}_\mathcal{B}}$, in the order of $l = 1, 2, \ldots, L$. Specifically, in the $l$-th layer, we first update the historical embedding of each node $v_i \in \mathcal{V}_\mathcal{B}$ as

$$\overline{\mathbf{h}}^l_i = u_{\theta^l}(\overline{\mathbf{h}}^{l-1}_i, \overline{\mathbf{m}}^{l-1}_{\mathcal{N}(v_i)}, \mathbf{x}_i);$$
$$\overline{\mathbf{m}}^{l-1}_{\mathcal{N}(v_i)} = \oplus_{\theta^l}\left(\left\{g_{\theta^l}(\overline{\mathbf{h}}^{l-1}_j) \mid v_j \in \mathcal{N}(v_i) \cap \mathcal{V}_\mathcal{B}\right\} \cup \left\{g_{\theta^l}(\widehat{\mathbf{h}}^{l-1}_j) \mid v_j \in \mathcal{N}(v_i) \setminus \mathcal{V}_\mathcal{B}\right\}\right). \quad (8)$$

Then, we compute the temporary embedding of each neighbor $v_i \in \mathcal{N}(\mathcal{V}_\mathcal{B}) \setminus \mathcal{V}_\mathcal{B}$ as

$$\widehat{\mathbf{h}}^l_i = (1 - \beta_i)\overline{\mathbf{h}}^l_i + \beta_i\widetilde{\mathbf{h}}^l_i, \quad (9)$$

where $\beta_i \in [0, 1]$ is the convex combination coefficient for node $v_i$, and

$$\widetilde{\mathbf{h}}^l_i = u_{\theta^l}(\widehat{\mathbf{h}}^{l-1}_i, \overline{\mathbf{m}}^{l-1}_{\mathcal{N}(v_i)}, \mathbf{x}_i);$$
$$\overline{\mathbf{m}}^{l-1}_{\mathcal{N}(v_i)} = \oplus_{\theta^l}\left(\left\{g_{\theta^l}(\overline{\mathbf{h}}^{l-1}_j) \mid v_j \in \mathcal{N}(v_i) \cap \mathcal{V}_\mathcal{B}\right\} \cup \left\{g_{\theta^l}(\widehat{\mathbf{h}}^{l-1}_j) \mid v_j \in \mathcal{N}(\mathcal{V}_\mathcal{B}) \cap \mathcal{N}(v_i) \setminus \mathcal{V}_\mathcal{B}\right\}\right).$$
$$(10)$$

We call $\mathbf{C}^l_f \triangleq \oplus_{\theta^l}\left(\left\{g_{\theta^l}(\widehat{\mathbf{h}}^{l-1}_j) \mid v_j \in \mathcal{N}(v_i) \setminus \mathcal{V}_\mathcal{B}\right\}\right)$ the *local message compensation* in the $l$-th layer in forward passes. For $l \in [L]$, $\overline{\mathbf{h}}^l_i$ is an approximation to $\mathbf{h}^l_i$ computed by Equation (2). Notice that the total size of Equations (8)–(10) is linear with $|\mathcal{N}(\mathcal{V}_\mathcal{B})|$ rather than the size of the whole graph. Suppose that the maximum neighborhood size is $n_{\max}$ and the number of layers is $L$, then the time complexity in forward passes is $\mathcal{O}(L(n_{\max}|\mathcal{V}_\mathcal{B}|d + |\mathcal{V}_\mathcal{B}|d^2))$.

In backward passes, we initialize the *temporary auxiliary variables* for $l = L$ as $\widehat{\mathbf{V}}^L = \nabla_\mathbf{H}\mathcal{L}$ and update historical auxiliary variables of nodes in $\mathcal{V}_\mathcal{B}$, i.e., $\overline{\mathbf{V}}^l_{\mathcal{V}_\mathcal{B}}$, in the order of $l = L - 1, \ldots, 1$.

Specifically, in the $l$-th layer, we first update the historical auxiliary variable of each $v_i \in \mathcal{V}_{\mathcal{B}}$ as

$$\overline{\mathbf{V}}_i^l = \sum_{v_j \in \mathcal{N}(v_i) \cap \mathcal{V}_{\mathcal{B}}} \left( \nabla_{\mathbf{h}_i^l} u_{\theta^{l+1}} (\overline{\mathbf{h}}_j^l, \overline{\mathbf{m}}_{\mathcal{N}(v_j)}^l, \mathbf{x}_j) \right) \overline{\mathbf{V}}_j^{l+1}$$
$$+ \sum_{v_j \in \mathcal{N}(v_i) \setminus \mathcal{V}_{\mathcal{B}}} \left( \nabla_{\mathbf{h}_i^l} u_{\theta^{l+1}} (\widehat{\mathbf{h}}_j^l, \overline{\mathbf{m}}_{\mathcal{N}(v_j)}^l, \mathbf{x}_j) \right) \widehat{\mathbf{V}}_j^{l+1}, \tag{11}$$

where $\overline{\mathbf{h}}_j^l$, $\overline{\mathbf{m}}_{\mathcal{N}(v_j)}^l$, and $\widehat{\mathbf{h}}_j^l$ are computed as shown in Equations (8)–(10). Then, we compute the temporary auxiliary variable of each neighbor $v_i \in \mathcal{N}(\mathcal{V}_{\mathcal{B}}) \setminus \mathcal{V}_{\mathcal{B}}$ as

$$\widehat{\mathbf{V}}_i^l = (1 - \beta_i) \overline{\mathbf{V}}_i^l + \beta_i \widetilde{\mathbf{V}}_i^l, \tag{12}$$

where $\beta_i$ is the convex combination coefficient used in Equation (9), and

$$\widetilde{\mathbf{V}}_i^l = \sum_{v_j \in \mathcal{N}(v_i) \cap \mathcal{V}_{\mathcal{B}}} \left( \nabla_{\mathbf{h}_i^l} u_{\theta^{l+1}} (\overline{\mathbf{h}}_j^l, \overline{\mathbf{m}}_{\mathcal{N}(v_j)}^l, \mathbf{x}_j) \right) \overline{\mathbf{V}}_j^{l+1}$$
$$+ \sum_{v_j \in \mathcal{N}(\mathcal{V}_{\mathcal{B}}) \cap \mathcal{N}(v_i) \setminus \mathcal{V}_{\mathcal{B}}} \left( \nabla_{\mathbf{h}_i^l} u_{\theta^{l+1}} (\widehat{\mathbf{h}}_j^l, \overline{\mathbf{m}}_{\mathcal{N}(v_j)}^l, \mathbf{x}_j) \right) \widehat{\mathbf{V}}_j^{l+1}. \tag{13}$$

We call $\mathbf{C}_b^l \triangleq \sum_{v_j \in \mathcal{N}(v_i) \setminus \mathcal{V}_{\mathcal{B}}} \left( \nabla_{\mathbf{h}_i^l} u_{\theta^{l+1}} (\widehat{\mathbf{h}}_j^l, \overline{\mathbf{m}}_{\mathcal{N}(v_j)}^l, \mathbf{x}_j) \right) \widehat{\mathbf{V}}_j^{l+1}$ the *local message compensation* in the $l$-th layer in backward passes. For $l \in [L]$, $\overline{\mathbf{V}}_i^l$ is an approximation to $\mathbf{V}_i^l$ computed by Equation (3). Similar to forward passes, the time complexity in backward passes is $\mathcal{O}(L(n_{\max}|\mathcal{V}_{\mathcal{B}}|d + |\mathcal{V}_{\mathcal{B}}|d^2))$, where $n_{\max}$ is the maximum neighborhood size and $L$ is the number of layers.

LMC additionally stores the historical node embeddings $\overline{\mathbf{H}}^l$ and auxiliary variables $\overline{\mathbf{V}}^l$ for $l \in [L]$. As pointed out in (Fey et al., 2021), we can store the majority of historical values in RAM or hard drive storage rather than GPU memory. Thus, the active historical values in forward and backward passes employ $\mathcal{O}(n_{\max}L|\mathcal{V}_{\mathcal{B}}|d)$ and $\mathcal{O}(n_{\max}L|\mathcal{V}_{\mathcal{B}}|d)$ GPU memory, respectively (see Appendix B). As the time and memory complexity are independent of the size of the whole graph, i.e., $|\mathcal{V}|$, LMC is scalable. We summarize the computational complexity in Appendix B.

Figure 1 shows the message passing mechanisms of GAS (Fey et al., 2021) and LMC. Compared with GAS, LMC proposes compensation messages between in-batch nodes and their 1-hop neighbors simultaneously in forward and backward passes. This corrects the biases of mini-batch gradients and thus accelerates convergence.

---

**Algorithm 1** Local Message Compensation

1: **Input:** The learning rate $\eta$ and the convex combination coefficients $(\beta_i)_{i=1}^n$.
2: Partition $\mathcal{V}$ into $B$ parts $(\mathcal{V}_b)_{b=1}^B$
3: **for** $k = 1, \ldots, N$ **do**
4:      Randomly sample $\mathcal{V}_{b_k}$ from $(\mathcal{V}_b)_{b=1}^B$
5:      Initialize $\overline{\mathbf{H}}^{0,k} = \widehat{\mathbf{H}}^{0,k} = \mathbf{X}$
6:      **for** $l = 1, \ldots, L$ **do**
7:          Update $\overline{\mathbf{H}}_{\mathcal{V}_{b_k}}^{l,k}$      ▷ (8)
8:          Compute $\widehat{\mathbf{H}}_{\mathcal{N}(\mathcal{V}_{b_k}) \setminus \mathcal{V}_{b_k}}^{l,k}$    ▷ (9) and (10)
9:      **end for**
10:     Initialize $\overline{\mathbf{V}}^{L,k} = \widehat{\mathbf{V}}^{L,k} = \nabla_{\mathbf{H}^L} \mathcal{L}$
11:     **for** $l = L - 1, \ldots, 1$ **do**
12:         Update $\overline{\mathbf{V}}_{\mathcal{V}_{b_k}}^{l,k}$       ▷ (11)
13:         Compute $\widehat{\mathbf{V}}_{\mathcal{N}(\mathcal{V}_{b_k}) \setminus \mathcal{V}_{b_k}}^{l,k}$   ▷ (12) and (13)
14:     **end for**
15:     Compute $\widetilde{\mathbf{g}}_w^k$ and $\widetilde{\mathbf{g}}_{\theta^l}^k$, $l \in [L]$    ▷ (6) and (7)
16:     Update parameters by
17:         $w^k = w^{k-1} - \eta \widetilde{\mathbf{g}}_w^k$
18:         $\theta^{l,k} = \theta^{l,k-1} - \eta \widetilde{\mathbf{g}}_{\theta^l}^k$, $l \in [L]$
19: **end for**

---

Algorithm 1 summarizes LMC. Unlike above, we add a superscript $k$ for each value to indicate that it is the value at the $k$-th iteration. At preprocessing step, we partition $\mathcal{V}$ into $B$ parts $(\mathcal{V}_b)_{b=1}^B$. At the $k$-th training step, LMC first randomly samples a subgraph constructed by $\mathcal{V}_{b_k}$. Notice that we sample more subgraphs to build a large graph in experiments whose convergence analysis is consistent with that of sampling a single subgraph. Then, LMC updates the stored historical node embeddings $\overline{\mathbf{H}}_{\mathcal{V}_{b_k}}^{l,k}$ in the order of $l = 1, \ldots, L$ by Equations (8)–(10), and the stored historical auxiliary variables $\overline{\mathbf{V}}_{\mathcal{V}_{b_k}}^{l,k}$ in the order of $l = L - 1, \ldots, 1$ by Equations (11)–(13). By the randomly updating, the historical values get close to the exact up-to-date values. Finally, for $l \in [L]$ and $v_j \in \mathcal{V}_{b_k}$, by replacing $\mathbf{h}_j^{l,k}$, $\mathbf{m}_{\mathcal{N}(v_j)}^{l,k}$ and $\mathbf{V}_j^{l,k}$ in Equations (6) and (7) with $\overline{\mathbf{h}}_j^{l,k}$, $\overline{\mathbf{m}}_{\mathcal{N}(v_j)}^{l,k}$, and $\overline{\mathbf{V}}_j^{l,k}$, respectively, LMC computes mini-batch gradients $\widetilde{\mathbf{g}}_w, \widetilde{\mathbf{g}}_{\theta^1}, \ldots, \widetilde{\mathbf{g}}_{\theta^L}$ to update parameters $w, \theta^1, \ldots, \theta^L$.

## 6 THEORETICAL ANALYSIS

In this section, we provide the theoretical analysis of LMC. Theorem 2 shows that the biases of mini-batch gradients computed by LMC can tend to an arbitrarily small value by setting a proper learning rate and convex combination coefficients. Then, Theorem 3 shows that LMC converges to first-order stationary points of GNNs. We provide detailed proofs of the theorems in Appendix D. In the theoretical analysis, we suppose that the following assumptions hold in this paper.

**Assumption 1.** *Assume that (1) at the $k$-th iteration, a batch of nodes $\mathcal{V}_{\mathcal{B}}^k$ is uniformly sampled from $\mathcal{V}$ and the corresponding labeled node set $\mathcal{V}_{L_{\mathcal{B}}}^k = \mathcal{V}_{\mathcal{B}}^k \cap \mathcal{V}_L$ is uniformly sampled from $\mathcal{V}_L$, (2) functions $f_{\theta^l}$, $\phi_{\theta^l}$, $\nabla_w \mathcal{L}$, $\nabla_{\theta^l} \mathcal{L}$, $\nabla_w \ell_w$, and $\nabla_{\theta^l} u_{\theta^l}$ are $\gamma$-Lipschitz with $\gamma > 1$, $\forall l \in [L]$, (3) norms $\|\mathbf{H}^{l,k}\|_F$, $\|\overline{\mathbf{H}}^{l,k}\|_F$, $\|\widehat{\mathbf{H}}^{l,k}\|_F$, $\|\widetilde{\mathbf{H}}^{l,k}\|_F$, $\|\mathbf{V}^{l,k}\|_F$, $\|\overline{\mathbf{V}}^{l,k}\|_F$, $\|\widehat{\mathbf{V}}^{l,k}\|_F$, $\|\widetilde{\mathbf{V}}^{l,k}\|_F$, $\|\nabla_w \mathcal{L}\|_2$, $\|\nabla_{\theta^l} \mathcal{L}\|_2$, $\|\widetilde{\mathbf{g}}_{\theta^l}\|_2$, and $\|\widetilde{\mathbf{g}}_w\|_2$ are bounded by $G > 1$, $\forall l \in [L]$, $k \in \mathbb{N}^*$.*

**Theorem 2.** *Suppose that Assumption 1 holds, then with $\eta = \mathcal{O}(\varepsilon^2)$ and $\beta_i = \mathcal{O}(\varepsilon^2)$, $i \in [n]$, there exists $C > 0$ and $\rho \in (0, 1)$ such that*

$$\mathbb{E}[\|\widetilde{\mathbf{g}}_w(w^k) - \nabla_w \mathcal{L}(w^k)\|_2] \leq C\varepsilon + C\rho^{\frac{k-1}{2}} + \mathrm{Var}(\mathbf{g}_w(w^k))^{\frac{1}{2}}, \ \forall k \in \mathbb{N}^*,$$

$$\mathbb{E}[\|\widetilde{\mathbf{g}}_{\theta^l}(\theta^{l,k}) - \nabla_{\theta^l} \mathcal{L}(\theta^{l,k})\|_2] \leq C\varepsilon + C\rho^{\frac{k-1}{2}} + \mathrm{Var}(\mathbf{g}_{\theta^l}(\theta^{l,k}))^{\frac{1}{2}}, \ \forall l \in [L], \ k \in \mathbb{N}^*.$$

**Theorem 3.** *Suppose that Assumption 1 holds. Besides, assume that the optimal value $\mathcal{L}^* = \inf_{w,\Theta} \mathcal{L}(w, \Theta)$ is bounded by $G$. Then, with $\eta = \mathcal{O}(\varepsilon^4)$, $\beta_i = \mathcal{O}(\varepsilon^4)$, $i \in [n]$, and $N = \mathcal{O}(\varepsilon^{-6})$, LMC ensures to find an $\varepsilon$-stationary solution such that $\mathbb{E}[\|\nabla_{w,\Theta} \mathcal{L}(w^R, \Theta^R)\|_2] \leq \varepsilon$ after running for $N$ iterations, where $R$ is uniformly selected from $[N]$ and $\Theta^R = (\theta^{l,R})_{l=1}^L$.*

## 7 EXPERIMENTS

We introduce experimental settings in Section 7.1. We then evaluate the convergence and efficiency of LMC in Sections 7.2 and 7.3. Finally, we conduct ablation studies about the proposed compensations in Section 7.4. We run all experiments on a single GeForce RTX 2080 Ti (11 GB).

### 7.1 EXPERIMENTAL SETTINGS

**Datasets.** Some recent works (Hu et al., 2020) have indicated that many frequently-used graph datasets are too small compared with graphs in real-world applications. Therefore, we evaluate LMC on four large datasets, PPI, REDDIT , FLICKR(Hamilton et al., 2017), and Ogbn-arxiv (Hu et al., 2020). These datasets contain thousands or millions of nodes/edges and have been widely used in previous works (Fey et al., 2021; Zeng et al., 2020; Hamilton et al., 2017; Chiang et al., 2019; Chen et al., 2018a;b). For more details, please refer to Appendix A.1.

**Baselines and Implementation Details.** In terms of prediction performance, our baselines include node-wise sampling methods (GraphSAGE (Hamilton et al., 2017) and VR-GCN (Chen et al., 2018a)), layer-wise sampling method (FASTGCN (Chen et al., 2018b) and LADIES (Zou et al., 2019)), subgraph-wise sampling methods (CLUSTER-GCN (Chiang et al., 2019), GRAPHSAINT (Zeng et al., 2020), FM (Yu et al., 2022), and GAS (Fey et al., 2021)), and a precomputing method (SIGN (Rossi et al., 2020)). By noticing that GAS and FM achieve the state-of-the-art prediction performance (Table 1) among the baselines, we further compare the efficiency of LMC with GAS, FM, and CLUSTER-GCN, another subgraph-wise sampling method using METIS partition. We implement LMC, FM, and CLUSTER-GCN based on the codes and toolkits of GAS (Fey et al., 2021) to ensure a fair comparison. For other implementation details, please refer to Appendix A.3.

**Hyperparameters.** To ensure a fair comparison, we follow the data splits, training pipeline, and most hyperparameters in (Fey et al., 2021) except for the additional hyperparameters in LMC such as $\beta_i$. We use the grid search to find the best $\beta_i$ (see Appendix A.4 for more details).

### 7.2 LMC IS FAST WITHOUT SACRIFICING ACCURACY

Table 1 reports the prediction performance of LMC and the baselines. We report the mean and the standard deviation by running each experiment five times for GAS, FM, and LMC. LMC, FM, and GAS all resemble full-batch performance on all datasets while other baselines may fail, especially on the FLICKR dataset. Moreover, LMC, FM, and GAS with deep GNNs, i.e., GCNII (Chen et al., 2020) outperform other baselines on all datasets.

As LMC, FM, and GAS share the similar prediction performance, we additionally compare the convergence speed of LMC, FM, GAS, and CLUSTER-GCN, another subgraph-wise sampling method using METIS partition, in Figure 2 and Table 2. We use a sliding window to smooth the convergence curve in Figure 2 as the accuracy on test data is unstable. The solid curves correspond to the mean, and the shaded regions correspond to values within plus or minus one standard deviation of the mean. Table 2 reports the number of epochs, the runtime to reach the full-batch accuracy in Table 1, and the GPU memory. As shown in Table 2 and Figure 2a, LMC is significantly faster than GAS, especially with a speed-up of 2x on REDDIT. Notably, the test accuracy of LMC is more stable than GAS, and thus the smooth test accuracy of LMC outperforms GAS in Figure 2b. Although GAS finally resembles full-batch performance

Table 1: **Prediction performance on large graph datasets.** OOM denotes the out-of-memory issue. Bold font indicates the best result and underline indicates the second best result.

| | **# nodes**
**# edges**
**Method** | 230K
11.6M
REDDIT | 57K
794K
PPI | 89K
450K
FLICKR | 169K
1.2M
ogbn-arxiv |
|---|---|---|---|---|---|
| | GRAPHSAGE | 95.40 | 61.20 | 50.10 | 71.49 |
| | VR-GCN | 94.50 | 85.60 | — | — |
| | FASTGCN | 93.70 | — | 50.40 | — |
| | LADIES | 92.80 | — | — | — |
| | CLUSTER-GCN | 96.60 | 99.36 | 48.10 | — |
| | GRAPHSAINT | 97.00 | **99.50** | 51.10 | — |
| | SIGN | 96.80 | 97.00 | 51.40 | — |
| GD | GCN | 95.43 | 97.58 | 53.73 | 71.64 |
| | GCNII | OOM | OOM | 55.28 | **72.83** |
| GAS | GCN | 95.35$_{\pm0.01}$ | 98.91$_{\pm0.03}$ | 53.44$_{\pm0.11}$ | 71.54$_{\pm0.19}$ |
| | GCNII | 96.73$_{\pm0.04}$ | 99.36$_{\pm0.02}$ | **55.42**$_{\pm\mathbf{0.27}}$ | 72.50$_{\pm0.28}$ |
| FM | GCN | 95.27$_{\pm0.03}$ | 98.91$_{\pm0.01}$ | 53.48$_{\pm0.17}$ | 71.49$_{\pm0.33}$ |
| | GCNII | 96.52$_{\pm0.06}$ | 99.34$_{\pm0.03}$ | 54.68$_{\pm0.27}$ | 72.54$_{\pm0.27}$ |
| LMC | GCN | 95.44$_{\pm0.02}$ | 98.87$_{\pm0.04}$ | 53.80$_{\pm0.14}$ | 71.44$_{\pm0.23}$ |
| | GCNII | **96.88**$_{\pm\mathbf{0.03}}$ | 99.32$_{\pm0.01}$ | 55.36$_{\pm0.49}$ | 72.76$_{\pm0.22}$ |

in Table 1 by selecting the best performance on the valid data, it may fail to resemble under small batch sizes due to its unstable process (see Section 7.3). Another appealing feature of LMC is that LMC shares comparable GPU memory costs with GAS, and thus LMC avoids the neighbor explosion problem. FM is slower than other methods, as they additionally update historical embeddings in the storage for the nodes outside the mini-batches. Please see Appendix E.2 for the comparison in terms of training time per epoch.

Table 2: Efficiency of CLUSTER-GCN, GAS, FM, and LMC.

| **Dataset & GNN** | **Epochs** | | | | **Runtime** (s) | | | | **Memory** (MB) | | | |
|---|---|---|---|---|---|---|---|---|---|---|---|---|
| | CLUSTER | GAS | FM | LMC | CLUSTER | GAS | FM | LMC | CLUSTER | GAS | FM | LMC |
| Ogbn-arxiv & GCN | 211.0 | 176.0 | 152.4 | **124.4** | 108 | 79 | 115 | **55** | **424** | 452 | 460 | 557 |
| FLICKR & GCN | 379.2 | 389.4 | 400.0 | **334.2** | 127 | 117 | 181 | **85** | **310** | 375 | 380 | 376 |
| REDDIT & GCN | 239.0 | 372.4 | 400.0 | **166.8** | 516 | 790 | 2269 | **381** | **1193** | 1508 | 1644 | 1829 |
| PPI & GCN | 428.0 | 293.6 | **286.4** | 290.2 | 359 | 179 | 224 | 179 | 212 | 214 | 218 | 267 |
| Ogbn-arxiv & GCNII | — | 234.8 | 373.6 | **197.4** | — | 218 | 381 | **178** | — | **453** | 454 | 568 |
| FLICKR & GCNII | — | **352** | 400.0 | 356 | — | **465** | 576 | 475 | — | **396** | 402 | 468 |

To further illustrate the convergence of LMC, we compare the errors of mini-batch gradients computed by CLUSTER, GAS, and LMC. At epoch training step, we record the relative errors $\|\widetilde{\mathbf{g}}_{\theta^l} - \nabla_{\theta^l}\mathcal{L}\|_2 / \|\nabla_{\theta^l}\mathcal{L}\|_2$, where $\nabla_{\theta^l}\mathcal{L}$ is the full-batch gradient for the parameters $\theta^l$ at the $l$-th MP layer and the $\widetilde{\mathbf{g}}_{\theta^l}$ is a mini-batch gradient. To avoid the randomness of the full-batch gradient $\nabla_{\theta^l}\mathcal{L}$, we set the dropout rate as zero. We report average relative errors during training in Figure 3. LMC enjoys the smallest estimated errors in the experiments.

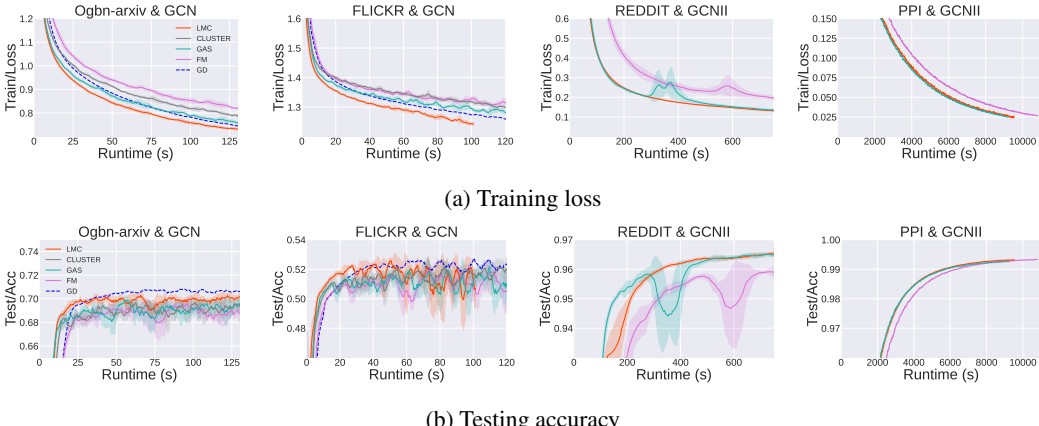

(a) Training loss

(b) Testing accuracy

Figure 2: Testing accuracy and training loss w.r.t. runtimes (s).

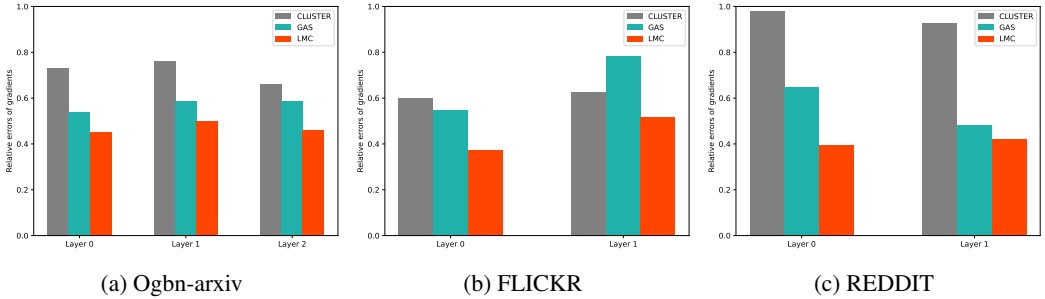

(a) Ogbn-arxiv      (b) FLICKR      (c) REDDIT

Figure 3: The average relative estimated errors of mini-batch gradients computed by CLUSTER, GAS, and LMC for GCN models.

### 7.3 LMC IS ROBUST IN TERMS OF BATCH SIZES

An appealing feature of mini-batch training methods is that they can avoid the out-of-memory issue by decreasing the batch size. Thus, we evaluate the prediction performance of LMC on Ogbn-arxiv datasets with different batch sizes (numbers of clusters). We conduct experiments under different sizes of sampled clusters per mini-batch. We run each experiment with the same epoch and search learning rates in the same

Table 3: Performance under different batch sizes on the Ogbn-arxiv dataset.

| Batch size | GCN | | GCNII | |
| --- | --- | --- | --- | --- |
| | GAS | LMC | GAS | LMC |
| 1 | 70.56 | **71.65** | 71.34 | **72.11** |
| 2 | 71.11 | **71.89** | 72.25 | **72.55** |
| 5 | **71.99** | 71.84 | 72.23 | **72.87** |
| 10 | 71.60 | **72.14** | **72.82** | 72.80 |

set. We report the best prediction accuracy in Table 3. LMC outperforms GAS under small batch sizes (batch size = 1 or 2) and achieve comparable performance with GAS (batch size = 5 or 10).

### 7.4 ABLATION

The improvement of LMC is due to two parts: the compensation in forward passes $\mathbf{C}_f^l$ and the compensation in back passes $\mathbf{C}_b^l$. Compared with GAS, the compensation in forward passes $\mathbf{C}_f^l$ additionally combines the incomplete up-to-date messages. Figure 4 shows the convergence curves of LMC using both $\mathbf{C}_f^l$ and $\mathbf{C}_b^l$ (denoted by $\mathbf{C}_f \& \mathbf{C}_b$), LMC using only $\mathbf{C}_f^l$ (denoted by $\mathbf{C}_f$), and GAS on the Ogbn-arxiv dataset. Under small batch sizes, the improvement mainly is due to $\mathbf{C}_b^l$ and the incomplete up-to-date messages in forward passes may hurt the performance. This is because the mini-batch and the union of their neighbors are hard to contain most neighbors of out-of-batch nodes when the batch size is small. Thus, the compensation in back passes $\mathbf{C}_b^l$ is the most important component by correcting the bias of the mini-batch gradients. Under large batch sizes, the improvement is due to $\mathbf{C}_f^l$, as the large batch sizes decrease the discarded messages and improve the accuracy of the mini-batch gradients (see Table 7 in Appendix). Notably, $\mathbf{C}_b^l$ still slightly improves the performance. We provide more ablation studies about $\beta_i$ in Appendix E.4.

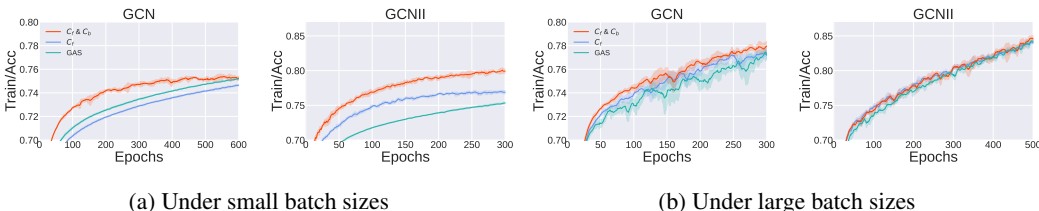

(a) Under small batch sizes      (b) Under large batch sizes

Figure 4: The improvement of the compensations on the Ogbn-arxiv dataset.

## 8 CONCLUSION

In this paper, we propose a novel subgraph-wise sampling method with a convergence guarantee, namely Local Message Compensation (LMC). LMC uses efficient and effective compensations to correct the biases of mini-batch gradients and thus accelerates convergence. We show that LMC converges to first-order stationary points of GNNs. To the best of our knowledge, LMC is the first subgraph-wise sampling method for GNNs with provable convergence. Experiments on large-scale benchmark tasks demonstrate that LMC significantly outperforms state-of-the-art subgraph-wise sampling methods in terms of efficiency.

## ACKNOWLEDGEMENT

The authors would like to thank all the anonymous reviewers for their insightful comments. This work was supported in part by National Nature Science Foundations of China grants U19B2026, U19B2044, 61836011, 62021001, and 61836006, and the Fundamental Research Funds for the Central Universities grant WK3490000004.

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

# A    More Details about Experiments

In this section, we introduce more details about our experiments, including datasets, training and evaluation protocols, and implementations.

## A.1    Datasets

We evaluate LMC on four large datasets, PPI, REDDIT, FLICKR (Hamilton et al., 2017), and Ogbn-arxiv (Hu et al., 2020). All of the datasets do not contain personally identifiable information or offensive content. Table 4 shows the summary statistics of the datasets. Details about the datasets are as follows.

- PPI contains 24 protein-protein interaction graphs. Each graph corresponds to a human tissue. Each node indicates a protein with positional gene sets, motif gene sets and immunological signatures as node features. Edges represent interactions between proteins. The task is to classify protein functions.
- REDDIT is a post-to-post graph constructed from REDDIT. Each node indicates a post and each edge between posts indicates that the same user comments on both. The task is to classify REDDIT posts into different communities based on (1) the GloVe CommonCrawl word vectors (Pennington et al., 2014) of the post titles and comments, (2) the post's scores, and (3) the number of comments made on the posts.
- Ogbn-arxiv is a directed citation network between all Computer Science (CS) arXiv papers indexed by MAG (Wang et al., 2020). Each node is an arXiv paper and each directed edge indicates that one paper cites another one. The task is to classify unlabeled arXiv papers into different primary categories based on labeled papers and node features, which are computed by averaging word2vec (Mikolov et al., 2013) embeddings of words in papers' title and abstract.
- FLICKR categorizes types of images based on their descriptions and properties (Fey et al., 2021; Zeng et al., 2020).

Table 4: Statistics of the datasets used in our experiments.

| Dataset | #Graphs | #Classes | Total #Nodes | Total #Edges |
|---|---|---|---|---|
| PPI | 24 | 121 | 56,944 | 793,632 |
| REDDIT | 1 | 41 | 232,965 | 11,606,919 |
| Ogbn-arxiv | 1 | 40 | 169,343 | 1,157,799 |
| FLICKR | 1 | 7 | 89,250 | 449,878 |

## A.2    Training and Evaluation Protocols

We run all the experiments on a single GeForce RTX 2080 Ti (11 GB). All the models are implemented in Pytorch (Paszke et al., 2019) and PyTorch Geometric (Fey & Lenssen, 2019) based on the official implementation of (Fey et al., 2021)[1]. The code of LMC is available on GitHub at `https://github.com/MIRALab-USTC/GNN-LMC`.

**Data Splitting.**    We use the data splitting strategies following previous works (Fey et al., 2021; Gu et al., 2020).

## A.3    Implementation Details

### A.3.1    Normalization Technique

In Section 4.2 in the main text, we assume that the subgraph $\mathcal{V}_\mathcal{B}$ is uniformly sampled from $\mathcal{V}$ and the corresponding set of labeled nodes $\mathcal{V}_{L_\mathcal{B}} = \mathcal{V}_\mathcal{B} \cap \mathcal{V}_L$ is uniformly sampled from $\mathcal{V}_L$. To enforce

---

[1]`https://github.com/rusty1s/pyg_autoscale`. The owner does not mention the license.

the assumption, we use the normalization technique to reweight Equations (6) and (7) in the main text.

Suppose we partition the whole graph $\mathcal{V}$ into $b$ parts $\{\mathcal{V}_{\mathcal{B}_i}\}_{i=1}^b$ and then uniformly sample $c$ clusters without replacement to construct subgraph $\mathcal{V}_\mathcal{B}$. By the normalization technique, Equation (6) becomes

$$\mathbf{g}_w(\mathcal{V}_\mathcal{B}) = \frac{b|\mathcal{V}_{L_\mathcal{B}}|}{c|\mathcal{V}_L|} \frac{1}{|\mathcal{V}_{L_\mathcal{B}}|} \sum_{v_j \in \mathcal{V}_{L_\mathcal{B}}} \nabla_w \ell_w(\mathbf{h}_j, y_j), \tag{14}$$

where $\frac{b|\mathcal{V}_{L_\mathcal{B}}|}{c|\mathcal{V}_L|}$ is the corresponding weight. Similarly, Equation (7) becomes

$$\mathbf{g}_\theta(\mathcal{V}_\mathcal{B}) = \frac{b|\mathcal{V}_\mathcal{B}|}{c|\mathcal{V}|} \frac{|\mathcal{V}|}{|\mathcal{V}_\mathcal{B}|} \sum_{v_j \in \mathcal{V}_\mathcal{B}} \nabla_\theta u(\mathbf{h}_j, \mathbf{m}_{\mathcal{N}(v_j)}, \mathbf{x}_j) \mathbf{V}_j, \tag{15}$$

where $\frac{b|\mathcal{V}_\mathcal{B}|}{c|\mathcal{V}|}$ is the corresponding weight.

### A.3.2 Incorporating Batch Normalization

We uniformly sample a mini-batch of nodes $\mathcal{V}_\mathcal{B}$ and generate the induced subgraph of $\mathcal{N}(\mathcal{V}_\mathcal{B})$. If we directly feed the $\mathbf{H}^{(l)}_{\mathcal{N}(\mathcal{V}_\mathcal{B})}$ to a batch normalization layer, the learned mean and standard deviation of the batch normalization layer may be biased. Thus, LMC first feeds the embeddings of the mini-batch $\mathbf{H}^{(l)}_{\mathcal{V}_\mathcal{B}}$ to a batch normalization layer and then feeds the embeddings outside the mini-batch $\mathbf{H}^{(l)}_{\mathcal{N}(\mathcal{V}_\mathcal{B}) \setminus \mathcal{V}_\mathcal{B}}$ to another batch normalization layer.

### A.4 Selection of $\beta_i$

We select $\beta_i = score(i)\alpha$ for each node $v_i$, where $\alpha \in [0, 1]$ is a hyperparameter and $score$ is a function to measure the quality of the incomplete up-to-date messages. We search $score$ in a $\{f(x) = x^2, f(x) = 2x - x^2, f(x) = x, f(x) = 1; x = deg_{local}(i)/deg_{global}(i)\}$, where $deg_{global}(i)$ is the degree of node $i$ in the whole graph and $deg_{local}(i)$ is the degree of node $i$ in the subgraph induced by $\mathcal{N}(\mathcal{V}_\mathcal{B})$.

## B Computational Complexity

We summarize the computational complexity in Table 5, where $n_{\max}$ is the maximum of neighborhoods, $L$ is the number of message passing layers, $\mathcal{V}_\mathcal{B}$ is a set of nodes in a sampled mini-batch, $d$ is the embedding dimension, $\mathcal{V}$ is the set of nodes in the whole graph, and $\mathcal{E}$ is the set of edges in the whole graph. As GD, backward SGD, CLUSTER, GAS, and LMC share the same memory complexity of parameters $\theta^{(l)}$, we omit it in Table 5.

Table 5: Time and memory complexity per gradient update of message passing based GNNs (e.g. GCN (Kipf & Welling, 2017) and GCNII (Chen et al., 2020)).

| Method | Time | Memory |
|---|---|---|
| GD and backward SGD | $\mathcal{O}(L(|\mathcal{E}|d + |\mathcal{V}|d^2))$ | $\mathcal{O}(L|\mathcal{V}|d)$ |
| CLUSTER (Chiang et al., 2019) | $\mathcal{O}(L(n_{\max}|\mathcal{V}_\mathcal{B}|d + |\mathcal{V}_\mathcal{B}|d^2))$ | $\mathcal{O}(L|\mathcal{V}_\mathcal{B}|d)$ |
| GAS (Fey et al., 2021) | $\mathcal{O}(L(n_{\max}|\mathcal{V}_\mathcal{B}|d + |\mathcal{V}_\mathcal{B}|d^2))$ | $\mathcal{O}(n_{\max}L|\mathcal{V}_\mathcal{B}|d)$ |
| LMC | $\mathcal{O}(L(n_{\max}|\mathcal{V}_\mathcal{B}|d + |\mathcal{V}_\mathcal{B}|d^2))$ | $\mathcal{O}(n_{\max}L|\mathcal{V}_\mathcal{B}|d)$ |

## C Additional Related Work

### C.1 Main differences between LMC and GraphFM

- First, LMC focuses on the convergence of subgraph-wise sampling methods, which is orthogonal to the idea of GraphFM-OB to alleviate the staleness problem of historical values. The

advanced approach to alleviating the staleness problem of historical values can further improve the performance of LMC and it is easy to establish provable convergence by the extension of LMC.

- Second, LMC uses nodes in both mini-batches and their 1-hop neighbors to compute incomplete up-to-date messages. In contrast, GraphFM-OB only uses nodes in the mini-batches. For the nodes whose neighbors are contained in the union of the nodes in mini-batches and their 1-hop neighbors, the aggregation results of LMC are exact, while those of GraphFM-OB are not.

- Third, by noticing that aggregation results are biased and the I/O bottleneck for the history access, LMC does not update the historical values in the storage for nodes outside the mini-batches. However, GraphFM-OB updates them based on the aggregation results.

## D DETAILED PROOFS

**Notations.** Unless otherwise specified, $C$ and $C'$ with any superscript or subscript denotes constants. We denote the learning rate by $\eta$.

In this section, we suppose that Assumption 1 holds.

### D.1 PROOF OF THEOREM 1: UNBIASED MINI-BATCH GRADIENTS OF BACKWARD SGD

In this subsection, we give the proof of Theorem 1, which shows that the mini-batch gradients computed by backward SGD are unbiased.

*Proof.* As $\mathcal{V}_{L_\mathcal{B}} = \mathcal{V}_\mathcal{B} \cap \mathcal{V}_L$ is uniformly sampled from $\mathcal{V}_L$, the expectation of $\mathbf{g}_w(\mathcal{V}_\mathcal{B})$ is

$$\mathbb{E}[\mathbf{g}_w(\mathcal{V}_\mathcal{B})] = \mathbb{E}[\frac{1}{|\mathcal{V}_{L_\mathcal{B}}|} \sum_{v_j \in \mathcal{V}_{L_\mathcal{B}}} \nabla_w \ell_w(\mathbf{h}_j, y_j)]$$
$$= \nabla_w \mathbb{E}[\ell_w(\mathbf{h}_j, y_j)]$$
$$= \nabla_w \mathcal{L}.$$

As the subgraph $\mathcal{V}_\mathcal{B}$ is uniformly sampled from $\mathcal{V}$, the expectation of $\mathbf{g}_{\theta^l}(\mathcal{V}_\mathcal{B})$ is

$$\mathbb{E}[\mathbf{g}_{\theta^l}(\mathcal{V}_\mathcal{B})] = \mathbb{E}[\frac{|\mathcal{V}|}{|\mathcal{V}_\mathcal{B}|} \sum_{v_j \in \mathcal{V}_\mathcal{B}} \left( \nabla_{\theta^l} u_{\theta^l}(\mathbf{h}_j^{l-1}, \mathbf{m}_{\mathcal{N}(v_j)}^{l-1}, \mathbf{x}_j) \right) \mathbf{V}_j^l]$$
$$= |\mathcal{V}| \mathbb{E}[\nabla_{\theta^l} u_{\theta^l}(\mathbf{h}_j^{l-1}, \mathbf{m}_{\mathcal{N}(v_j)}^{l-1}, \mathbf{x}_j) \mathbf{V}_j^l]$$
$$= |\mathcal{V}| \frac{1}{|\mathcal{V}|} \sum_{v_j \in \mathcal{V}} \nabla_{\theta^l} u_{\theta^l}(\mathbf{h}_j^{l-1}, \mathbf{m}_{\mathcal{N}(v_j)}^{l-1}, \mathbf{x}_j) \mathbf{V}_j^l$$
$$= \sum_{v_j \in \mathcal{V}} \nabla_{\theta^l} u_{\theta^l}(\mathbf{h}_j^{l-1}, \mathbf{m}_{\mathcal{N}(v_j)}^{l-1}, \mathbf{x}_j) \mathbf{V}_j^l$$
$$= \nabla_{\theta^l} \mathcal{L}, \ \forall l \in [L].$$

$\square$

### D.2 DIFFERENCES BETWEEN EXACT VALUES AT ADJACENT ITERATIONS

We first show that the differences between the exact values of the same layer in two adjacent iterations can be bounded by setting a proper learning rate.

**Lemma 1.** *Suppose that Assumption 1 holds. Given an L-layer GNN, for any $\varepsilon > 0$, by letting*

$$\eta \leq \frac{\varepsilon}{(2\gamma)^L G} < \varepsilon,$$

*we have*

$$\|\mathbf{H}^{l,k+1} - \mathbf{H}^{l,k}\|_F < \varepsilon, \ \forall l \in [L], k \in \mathbb{N}^*.$$

*Proof.* Since $\eta \leq \frac{\varepsilon}{(2\gamma)^L G} < \frac{\varepsilon}{\gamma(2\gamma)^{L-1} G}$, we have

$$
\begin{aligned}
\|\mathbf{H}^{1,k+1} - \mathbf{H}^{1,k}\|_F &= \|f_{\theta^{1,k+1}}(\mathbf{X}) - f_{\theta^{1,k}}(\mathbf{X})\|_F \\
&\leq \gamma \|\theta^{1,k+1} - \theta^{1,k}\| \\
&\leq \gamma \|\widetilde{\mathbf{g}}_{\theta^1}\| \eta \\
&< \frac{\gamma G \varepsilon}{\gamma(2\gamma)^{L-1} G} \\
&= \frac{\varepsilon}{(2\gamma)^{L-1}}.
\end{aligned}
$$

Then, because $\eta \leq \frac{\varepsilon}{(2\gamma)^L G} < \frac{\varepsilon}{(2\gamma)^{L-1} G}$, we have

$$
\begin{aligned}
\|\mathbf{H}^{2,k+1} - \mathbf{H}^{2,k}\|_F &= \|f_{\theta^{2,k+1}}(\mathbf{H}^{1,k+1}) - f_{\theta^{2,k}}(\mathbf{H}^{1,k})\|_F \\
&\leq \|f_{\theta^{2,k+1}}(\mathbf{H}^{1,k+1}) - f_{\theta^{2,k}}(\mathbf{H}^{1,k+1})\|_F + \|f_{\theta^{2,k}}(\mathbf{H}^{1,k+1}) - f_{\theta^{2,k}}(\mathbf{H}^{1,k})\|_F \\
&\leq \gamma \|\theta^{2,k+1} - \theta^{2,k}\| + \gamma \|\mathbf{H}^{1,k+1} - \mathbf{H}^{1,k}\|_F \\
&\leq \gamma G \eta + \frac{\varepsilon}{2(2\gamma)^{L-2}} \\
&< \frac{\varepsilon}{2(2\gamma)^{L-2}} + \frac{\varepsilon}{2(2\gamma)^{L-2}} \\
&= \frac{\varepsilon}{(2\gamma)^{L-2}}.
\end{aligned}
$$

And so on, we have

$$
\|\mathbf{H}^{l,k+1} - \mathbf{H}^{l,k}\|_F < \frac{\varepsilon}{(2\gamma)^{L-l}}, \ \forall l \in [L], \ k \in \mathbb{N}^*.
$$

Since $(2\gamma)^{L-l} > 1$, we have

$$
\|\mathbf{H}^{l,k+1} - \mathbf{H}^{l,k}\|_F < \varepsilon, \ \forall l \in [L], \ k \in \mathbb{N}^*.
$$

$\square$

**Lemma 2.** *Suppose that Assumption 1 holds. Given an L-layer GNN, for any $\varepsilon > 0$, by letting*

$$
\eta \leq \frac{\varepsilon}{(2\gamma)^{L-1} G} < \varepsilon,
$$

*we have*

$$
\|\mathbf{V}^{l,k+1} - \mathbf{V}^{l,k}\|_F < \varepsilon, \ \forall l \in [L], \ k \in \mathbb{N}^*.
$$

*Proof.* Since $\eta \leq \frac{\varepsilon}{(2\gamma)^{L-1} G} < \frac{\varepsilon}{\gamma(2\gamma)^{L-2} G}$, we have

$$
\begin{aligned}
\|\mathbf{V}^{L-1,k+1} - \mathbf{V}^{L-1,k}\|_F &= \|\phi_{\theta^{L,k+1}}(\nabla_{\mathbf{H}}\mathcal{L}) - \phi_{\theta^{L,k}}(\nabla_{\mathbf{H}}\mathcal{L})\|_F \\
&\leq \gamma \|\theta^{L,k+1} - \theta^{L,k}\| \\
&\leq \gamma \|\widetilde{\mathbf{g}}_{\theta^L}\| \eta \\
&< \frac{\gamma G \varepsilon}{\gamma(2\gamma)^{L-2} G} \\
&= \frac{\varepsilon}{(2\gamma)^{L-2}}.
\end{aligned}
$$

Then, because $\eta \leq \frac{\varepsilon}{(2\gamma)^{L-1}G} < \frac{\varepsilon}{(2\gamma)^{L-2}G}$, we have

$$
\begin{aligned}
&\|\mathbf{V}^{L-2,k+1} - \mathbf{V}^{L-2,k}\|_F \\
&= \|\phi_{\theta^{L-1,k+1}}(\mathbf{V}^{L-1,k+1}) - \phi_{\theta^{L-1,k}}(\mathbf{V}^{L-1,k})\|_F \\
&\leq \|\phi_{\theta^{L-1,k+1}}(\mathbf{V}^{L-1,k+1}) - \phi_{\theta^{L-1,k}}(\mathbf{V}^{L-1,k+1})\|_F \\
&\quad + \|\phi_{\theta^{L-1,k}}(\mathbf{V}^{L-1,k+1}) - \phi_{\theta^{L-1,k}}(\mathbf{V}^{L-1,k})\|_F \\
&\leq \gamma\|\theta^{L-1,k+1} - \theta^{L-1,k}\| + \gamma\|\mathbf{V}^{L-1,k+1} - \mathbf{V}^{L-1,k}\|_F \\
&\leq \gamma G\eta + \frac{\varepsilon}{2(2\gamma)^{L-3}} \\
&< \frac{\varepsilon}{2(2\gamma)^{L-3}} + \frac{\varepsilon}{2(2\gamma)^{L-3}} \\
&= \frac{\varepsilon}{(2\gamma)^{L-3}}.
\end{aligned}
$$

And so on, we have

$$
\|\mathbf{V}^{l,k+1} - \mathbf{V}^{l,k}\|_F < \frac{\varepsilon}{(2\gamma)^{l-1}}, \ \forall l \in [L], \ k \in \mathbb{N}^*.
$$

Since $(2\gamma)^{l-1} > 1$, we have

$$
\|\mathbf{V}^{l,k+1} - \mathbf{V}^{l,k}\|_F < \varepsilon, \ \forall l \in [L], \ k \in \mathbb{N}^*.
$$

$\square$

### D.3 HISTORICAL VALUES AND TEMPORARY VALUES

Suppose that we uniformly sample a mini-batch $\mathcal{V}_{\mathcal{B}}^k \subset \mathcal{V}$ at the $k$-th iteration and $|\mathcal{V}_{\mathcal{B}}^k| = S$. For the simplicity of notations, we denote the temporary node embeddings and auxiliary variables in the $l$-th layer by $\widehat{\mathbf{H}}^{l,k}$ and $\widehat{\mathbf{V}}^{l,k}$, respectively, where

$$
\widehat{\mathbf{H}}_i^{l,k} = \begin{cases} \widehat{\mathbf{h}}_i^{l,k}, & v_i \in \mathcal{N}(\mathcal{V}_{\mathcal{B}}^k) \setminus \mathcal{V}_{\mathcal{B}}^k, \\ \overline{\mathbf{h}}_i^{l,k}, & \text{otherwise}, \end{cases}
$$

and

$$
\widehat{\mathbf{V}}_i^{l,k} = \begin{cases} \widehat{\mathbf{v}}_i^{l,k}, & v_i \in \mathcal{N}(\mathcal{V}_{\mathcal{B}}^k) \setminus \mathcal{V}_{\mathcal{B}}^k, \\ \overline{\mathbf{v}}_i^{l,k}, & \text{otherwise}. \end{cases}
$$

We abbreviate the process that LMC updates the node embeddings and auxiliary variables of $\mathcal{V}_{\mathcal{B}}^k$ in the $l$-th layer at the $k$-th iteration as

$$
\begin{aligned}
\overline{\mathbf{H}}_{\mathcal{V}_{\mathcal{B}}^k}^{l,k} &= [f_{\theta^{l,k}}(\widehat{\mathbf{H}}^{l-1,k})]_{\mathcal{V}_{\mathcal{B}}^k}, \\
\overline{\mathbf{V}}_{\mathcal{V}_{\mathcal{B}}^k}^{l,k} &= [\phi_{\theta^{l+1,k}}(\widehat{\mathbf{H}}^{l+1,k})]_{\mathcal{V}_{\mathcal{B}}^k}.
\end{aligned}
$$

For each $v_i \in \mathcal{V}_{\mathcal{B}}^k$, the update process of $v_i$ in the $l$-th layer at the $k$-th iteration can be expressed by

$$
\begin{aligned}
\overline{\mathbf{h}}_i^{l,k} &= f_{\theta^{l,k},i}(\widehat{\mathbf{H}}^{l-1,k}), \\
\overline{\mathbf{V}}_i^{l,k} &= \phi_{\theta^{l+1,k},i}(\widehat{\mathbf{V}}^{l+1,k}),
\end{aligned}
$$

where $f_{\theta^{l,k},i}$ and $\phi_{\theta^{l+1,k},i}$ are the components for node $v_i$ of $f_{\theta^{l,k}}$ and $\phi_{\theta^{l+1,k}}$, respectively.

#### D.3.1 CONVEX COMBINATION COEFFICIENTS

We first focus on convex combination coefficients $\beta_i$, $i \in [n]$. For the simplicity of analysis, we assume $\beta_i = \beta$ for $i \in [i]$. The analysis of the case where $(\beta_i)_{i=1}^n$ are different from each other is the same.

**Lemma 3.** *Suppose that Assumption 1 holds. For any $\varepsilon > 0$, by letting*

$$\beta \leq \frac{\varepsilon}{2G}, \ \forall l \in [L], \ i \in [n],$$

*we have*

$$\|\widehat{\mathbf{H}}^{l,k} - \mathbf{H}^{l,k}\|_F \leq \|\overline{\mathbf{H}}^{l,k} - \mathbf{H}^{l,k}\|_F + \varepsilon, \ \forall l \in [L], \ k \in \mathbb{N}^*.$$

*Proof.* Since $\widehat{\mathbf{H}}^{l,k} = (1 - \beta)\overline{\mathbf{H}}^{l,k} + \beta\widetilde{\mathbf{H}}^{l,k}$, we have

$$
\begin{aligned}
\|\widehat{\mathbf{H}}^{l,k} - \mathbf{H}^{l,k}\|_F &= \|(1 - \beta)\overline{\mathbf{H}}^{l,k} + \beta\widetilde{\mathbf{H}}^{l,k} - (1 - \beta)\mathbf{H}^{l,k} + \beta\mathbf{H}^{l,k}\|_F \\
&\leq (1 - \beta)\|\overline{\mathbf{H}}^{l,k} - \mathbf{H}^{l,k}\|_F + \beta\|\widetilde{\mathbf{H}}^{l,k} - \mathbf{H}^{l,k}\|_F \\
&\leq \|\overline{\mathbf{H}}^{l,k} - \mathbf{H}^{l,k}\|_F + 2\beta G.
\end{aligned}
$$

Hence letting $\beta \leq \frac{\varepsilon}{2G}$ leads to

$$\|\widehat{\mathbf{H}}^{l,k} - \mathbf{H}^{l,k}\|_F \leq \|\overline{\mathbf{H}}^{l,k} - \mathbf{H}^{l,k}\|_F + \varepsilon.$$

$\square$

**Lemma 4.** *Suppose that Assumption 1 holds. For any $\varepsilon > 0$, by letting*

$$\beta \leq \frac{\varepsilon}{2G}, \ \forall l \in [L], \ i \in [n],$$

*we have*

$$\|\widehat{\mathbf{V}}^{l,k} - \mathbf{V}^{l,k}\|_F \leq \|\overline{\mathbf{V}}^{l,k} - \mathbf{V}^{l,k}\|_F + \varepsilon.$$

*Proof.* Since $\widehat{\mathbf{H}}^{l,k} = (1 - \beta)\overline{\mathbf{H}}^{l,k} + \beta\widetilde{\mathbf{H}}^{l,k}$, we have

$$
\begin{aligned}
\|\widehat{\mathbf{H}}^{l,k} - \mathbf{H}^{l,k}\| &= \|(1 - \beta)\overline{\mathbf{H}}^{l,k} + \beta\widetilde{\mathbf{H}}^{l,k} - (1 - \beta)\mathbf{H}^{l,k} + \beta\mathbf{H}^{l,k}\|_F \\
&\leq (1 - \beta)\|\overline{\mathbf{H}}^{l,k} - \mathbf{H}^{l,k}\|_F + \beta\|\widetilde{\mathbf{H}}^{l,k} - \mathbf{H}^{l,k}\|_F \\
&\leq \|\overline{\mathbf{H}}^{l,k} - \mathbf{H}^{l,k}\|_F + 2\beta G.
\end{aligned}
$$

Hence letting $\beta \leq \frac{\varepsilon}{2G}$ leads to

$$\|\widehat{\mathbf{H}}^{l,k} - \mathbf{H}^{l,k}\|_F \leq \|\overline{\mathbf{H}}^{l,k} - \mathbf{H}^{l,k}\|_F + \varepsilon.$$

$\square$

### D.3.2 Approximation errors of historical values

Next, we focus on the approximation errors of historical node embeddings and auxiliary variables

$$d_h^{l,k} := \left(\mathbb{E}[\|\overline{\mathbf{H}}^{l,k} - \mathbf{H}^{l,k}\|_F^2]\right)^{\frac{1}{2}}, \ l \in [L],$$

$$d_v^{l,k} := \left(\mathbb{E}[\|\overline{\mathbf{V}}^{l,k} - \mathbf{V}^{l,k}\|_F^2]\right)^{\frac{1}{2}}, \ l \in [L - 1].$$

**Lemma 5.** *For an $L$-layer GNN, suppose that Assumption 1 holds. Besides, we suppose that*

1. *$(d_h^{l,1})^2$ is bounded by $G > 1$, $\forall l \in [L]$,*

2. *there exists $N \in \mathbb{N}^*$ such that*

$$\|\widehat{\mathbf{H}}^{l,k} - \mathbf{H}^{l,k}\|_F \leq \|\overline{\mathbf{H}}^{l,k} - \mathbf{H}^{l,k}\|_F + \frac{1}{N^{\frac{2}{3}}}, \ \forall l \in [L], \ k \in \mathbb{N}^*,$$

$$\|\mathbf{H}^{l,k} - \mathbf{H}^{l,k-1}\|_F \leq \frac{1}{N^{\frac{2}{3}}}, \ \forall k \in \mathbb{N}^*,$$

then there exist constants $C'_{*,1}$, $C'_{*,2}$, and $C'_{*,3}$ that do not depend on $k, l, N$, and $\eta$, such that

$$(d_h^{l,k+1})^2 \le C'_{*,1}\eta + C'_{*,2}\rho^k + \frac{C'_{*,3}}{N^{\frac{2}{3}}}, \ \forall l \in [L], \ k \in \mathbb{N}^*,$$

where $\rho = \frac{n-S}{n} < 1$, $n = |\mathcal{V}|$, and $S$ is number of sampled nodes at each iteration.

*Proof.* We have

$$(d_h^{l+1,k+1})^2$$
$$= \mathbb{E}[\|\overline{\mathbf{H}}^{l+1,k+1} - \mathbf{H}^{l+1,k+1}\|_F^2]$$
$$= \mathbb{E}[\sum_{i=1}^n \|\overline{\mathbf{h}}_i^{l+1,k+1} - \mathbf{h}_i^{l+1,k+1}\|_F^2]$$
$$= \mathbb{E}[\sum_{v_i \in \mathcal{V}_\mathcal{B}^k} \|f_{\theta^{l+1,k+1},i}(\widehat{\mathbf{H}}^{l,k+1}) - f_{\theta^{l+1,k+1},i}(\mathbf{H}^{l,k+1})\|_F^2$$
$$+ \sum_{v_i \notin \mathcal{V}_\mathcal{B}^k} \|\overline{\mathbf{h}}_i^{l+1,k} - f_{\theta^{l+1,k+1},i}(\mathbf{H}^{l,k+1})\|_F^2]$$
$$= \mathbb{E}[\frac{S}{n}\sum_{i=1}^n \|f_{\theta^{l+1,k+1},i}(\widehat{\mathbf{H}}^{l,k+1}) - f_{\theta^{l+1,k+1},i}(\mathbf{H}^{l,k+1})\|_F^2$$
$$+ \frac{n-S}{n}\sum_{i=1}^n \|\overline{\mathbf{h}}_i^{l+1,k} - f_{\theta^{l+1,k+1},i}(\mathbf{H}^{l,k+1})\|_F^2]$$
$$\le \frac{S}{n}\sum_{i=1}^n \mathbb{E}[\|f_{\theta^{l+1,k+1},i}(\widehat{\mathbf{H}}^{l,k+1}) - f_{\theta^{l+1,k+1},i}(\mathbf{H}^{l,k+1})\|_F^2]$$
$$+ \frac{n-S}{n}\sum_{i=1}^n \mathbb{E}[\|\overline{\mathbf{h}}_i^{l+1,k} - f_{\theta^{l+1,k+1},i}(\mathbf{H}^{l,k+1})\|_F^2].$$

About the first term, for $l \ge 1$, we have

$$\mathbb{E}[\|f_{\theta^{l+1,k+1},i}(\widehat{\mathbf{H}}^{l,k+1}) - f_{\theta^{l+1,k+1},i}(\mathbf{H}^{l,k+1})\|_F^2]$$
$$\le \gamma^2 \mathbb{E}[\|\widehat{\mathbf{H}}^{l,k+1} - \mathbf{H}^{l,k+1}\|_F^2]$$
$$\le \gamma^2 \mathbb{E}[(\|\overline{\mathbf{H}}^{l,k+1} - \mathbf{H}^{l,k+1}\|_F + \frac{1}{N^{\frac{2}{3}}})^2]$$
$$\le 2\gamma^2 \mathbb{E}[\|\overline{\mathbf{H}}^{l,k+1} - \mathbf{H}^{l,k+1}\|_F^2] + \frac{2\gamma^2}{N^{\frac{4}{3}}}$$
$$= 2\gamma^2 (d_h^{l,k+1})^2 + \frac{2\gamma^2}{N^{\frac{4}{3}}}.$$

For $l = 0$, we have

$$\mathbb{E}[\|f_{\theta^{l+1,k+1},i}(\widehat{\mathbf{H}}^{l,k+1}) - f_{\theta^{l+1,k+1},i}(\mathbf{H}^{l,k+1})\|_F^2]$$
$$= \mathbb{E}[\|f_{\theta^{1,k+1},i}(\widehat{\mathbf{H}}^{0,k+1}) - f_{\theta^{1,k+1},i}(\mathbf{H}^{0,k+1})\|_F^2]$$
$$= \mathbb{E}[\|f_{\theta^{1,k+1},i}(\mathbf{X}) - f_{\theta^{1,k+1},i}(\mathbf{X})\|_F^2]$$
$$= 0.$$

About the second term, for $l \geq 1$, we have

$$\mathbb{E}[\|\overline{\mathbf{h}}_i^{l+1,k} - f_{\theta^{l+1,k+1},i}(\mathbf{H}^{l,k+1})\|_F^2]$$

$$\leq \mathbb{E}[\|\overline{\mathbf{h}}_i^{l+1,k} - \mathbf{h}_i^{l+1,k} + \mathbf{h}_i^{l+1,k} - f_{\theta^{l+1,k+1},i}(\mathbf{H}^{l,k+1})\|_F^2]$$

$$\leq \mathbb{E}[\|\overline{\mathbf{h}}_i^{l+1,k} - \mathbf{h}_i^{l+1,k}\|_F^2] + \mathbb{E}[\|\mathbf{h}_i^{l+1,k} - f_{\theta^{l+1,k+1},i}(\mathbf{H}^{l,k+1})\|_F^2]$$
$$\quad + 2\mathbb{E}[\langle \overline{\mathbf{h}}_i^{l+1,k} - \mathbf{h}_i^{l+1,k}, \mathbf{h}_i^{l+1,k} - f_{\theta^{l+1,k+1},i}(\mathbf{H}^{l,k+1})\rangle]$$

$$\leq \mathbb{E}[\|\overline{\mathbf{h}}_i^{l+1,k} - \mathbf{h}_i^{l+1,k}\|_F^2]$$
$$\quad + \mathbb{E}[\|\mathbf{h}_i^{l+1,k} - f_{\theta^{l+1,k+1},i}(\mathbf{H}^{l,k}) + f_{\theta^{l+1,k+1},i}(\mathbf{H}^{l,k}) - f_{\theta^{l+1,k+1},i}(\mathbf{H}^{l,k+1})\|_F^2]$$
$$\quad + 24\mathbb{E}[\langle \overline{\mathbf{h}}_i^{l+1,k} - \mathbf{h}_i^{l+1,k}, \mathbf{h}_i^{l+1,k} - f_{\theta^{l+1,k+1},i}(\mathbf{H}^{l,k+1})\rangle]$$

$$\leq \mathbb{E}[\|\overline{\mathbf{h}}_i^{l+1,k} - \mathbf{h}_i^{l+1,k}\|_F^2]$$
$$\quad + 2\mathbb{E}[\|\mathbf{h}_i^{l+1,k} - f_{\theta^{l+1,k+1},i}(\mathbf{H}^{l,k})\|_F^2] + 2\mathbb{E}[\|f_{\theta^{l+1,k+1},i}(\mathbf{H}^{l,k}) - f_{\theta^{l+1,k+1},i}(\mathbf{H}^{l,k+1})\|_F^2]$$
$$\quad + 4G\mathbb{E}[\|\mathbf{h}_i^{l+1,k} - f_{\theta^{l+1,k+1},i}(\mathbf{H}^{l,k+1})\|_F]$$

$$\leq \mathbb{E}[\|\overline{\mathbf{h}}_i^{l+1,k} - \mathbf{h}_i^{l+1,k}\|_F^2]$$
$$\quad + 2\gamma^2\mathbb{E}[\|\theta^{l+1,k} - \theta^{l+1,k+1}\|^2] + 2\gamma^2\mathbb{E}[\|\mathbf{H}^{l,k} - \mathbf{H}^{l,k+1}\|_F^2]$$
$$\quad + 4G\gamma\mathbb{E}[\|\theta^{l+1,k} - \theta^{l+1,k+1}\| + \|\mathbf{H}^{l,k} - \mathbf{H}^{l,k+1}\|_F]$$

$$\leq \mathbb{E}[\|\overline{\mathbf{h}}_i^{l+1,k} - \mathbf{h}_i^{l+1,k}\|_F^2] + 2\gamma^2 G^2 \eta^2 + 4G^2\gamma\eta + \frac{2\gamma^2}{N^{\frac{4}{3}}} + \frac{4G\gamma}{N^{\frac{2}{3}}}$$

$$\leq \mathbb{E}[\|\overline{\mathbf{h}}_i^{l+1,k} - \mathbf{h}_i^{l+1,k}\|_F^2] + 2G^2\gamma(\gamma+2)\eta + \frac{2\gamma(\gamma+2G)}{N^{\frac{2}{3}}}.$$

For $l = 0$, we have

$$\mathbb{E}[\|\overline{\mathbf{h}}_i^{l+1,k} - f_{\theta^{l+1,k+1},i}(\mathbf{H}^{l,k+1})\|_F^2]$$

$$\leq \mathbb{E}[\|\overline{\mathbf{h}}_i^{l+1,k} - \mathbf{h}_i^{l+1,k} + \mathbf{h}_i^{l+1,k} - f_{\theta^{l+1,k+1},i}(\mathbf{H}^{l,k+1})\|_F^2]$$

$$\leq \mathbb{E}[\|\overline{\mathbf{h}}_i^{l+1,k} - \mathbf{h}_i^{l+1,k}\|_F^2] + \mathbb{E}[\|\mathbf{h}_i^{l+1,k} - f_{\theta^{l+1,k+1},i}(\mathbf{H}^{l,k+1})\|_F^2]$$
$$\quad + 2\mathbb{E}[\langle \overline{\mathbf{h}}_i^{l+1,k} - \mathbf{h}_i^{l+1,k}, \mathbf{h}_i^{l+1,k} - f_{\theta^{l+1,k+1},i}(\mathbf{H}^{l,k+1})\rangle]$$

$$\leq \mathbb{E}[\|\overline{\mathbf{h}}_i^{l+1,k} - \mathbf{h}_i^{l+1,k}\|_F^2]$$
$$\quad + \mathbb{E}[\|\mathbf{h}_i^{l+1,k} - f_{\theta^{l+1,k+1},i}(\mathbf{H}^{l,k}) + f_{\theta^{l+1,k+1},i}(\mathbf{H}^{l,k}) - f_{\theta^{l+1,k+1},i}(\mathbf{H}^{l,k+1})\|_F^2]$$
$$\quad + 2\mathbb{E}[\langle \overline{\mathbf{h}}_i^{l+1,k} - \mathbf{h}_i^{l+1,k}, \mathbf{h}_i^{l+1,k} - f_{\theta^{l+1,k+1},i}(\mathbf{H}^{l,k+1})\rangle]$$

$$\leq \mathbb{E}[\|\overline{\mathbf{h}}_i^{l+1,k} - \mathbf{h}_i^{l+1,k}\|_F^2]$$
$$\quad + 2\mathbb{E}[\|\mathbf{h}_i^{l+1,k} - f_{\theta^{l+1,k+1},i}(\mathbf{H}^{l,k})\|_F^2] + 2\mathbb{E}[\|f_{\theta^{l+1,k+1},i}(\mathbf{H}^{l,k}) - f_{\theta^{l+1,k+1},i}(\mathbf{H}^{l,k+1})\|_F^2]$$
$$\quad + 4G\mathbb{E}[\|\mathbf{h}_i^{l+1,k} - f_{\theta^{l+1,k+1},i}(\mathbf{H}^{l,k+1})\|_F]$$

$$\leq \mathbb{E}[\|\overline{\mathbf{h}}_i^{l+1,k} - \mathbf{h}_i^{l+1,k}\|_F^2] + 2\gamma^2\mathbb{E}[\|\theta^{l+1,k} - \theta^{l+1,k+1}\|^2] + 4G\gamma\mathbb{E}[\|\theta^{l+1,k} - \theta^{l+1,k+1}\|]$$

$$\leq \mathbb{E}[\|\overline{\mathbf{h}}_i^{l+1,k} - \mathbf{h}_i^{l+1,k}\|_F^2] + 2\gamma^2 G^2 \eta^2 + 4G^2\gamma\eta,$$

$$\leq \mathbb{E}[\|\overline{\mathbf{h}}_i^{l+1,k} - \mathbf{h}_i^{l+1,k}\|_F^2] + 2G^2\gamma(\gamma+2)\eta + 4G^2\gamma\eta.$$

Hence we have

$$(d_h^{l+1,k+1})^2 \leq \frac{(n-S)}{n}(d_h^{l+1,k})^2 + 2(n-S)\gamma(\gamma+2)G^2\eta$$
$$+ \begin{cases} 0, & l = 0, \\ 2\gamma^2 S(d_h^{l,k+1})^2 + \frac{4n\gamma(\gamma+G)}{N^{\frac{2}{3}}}, & l \geq 1. \end{cases}$$

Let $\rho = \frac{n-S}{n} < 1$. For $l = 0$, we have

$$(d_h^{1,k+1})^2 - \frac{2(n-S)\gamma(\gamma+2)G^2\eta}{1-\rho} \leq \rho((d_h^{1,k})^2 - \frac{2(n-S)\gamma(\gamma+2)G^2\eta}{1-\rho})$$
$$\leq \rho^2((d_h^{1,k-1})^2 - \frac{2(n-S)\gamma(\gamma+2)G^2\eta}{1-\rho})$$
$$\leq \cdots$$
$$\leq \rho^k((d_h^{1,1})^2 - \frac{2(n-S)\gamma(\gamma+2)G^2\eta}{1-\rho})$$
$$\leq \rho^k G,$$

which leads to

$$(d_h^{1,k+1})^2 \leq \frac{2(n-S)\gamma(\gamma+2)G^2}{1-\rho}\eta + \rho^k G = C'_{1,1}\eta + \rho^k G.$$

Then, for $l = 1$ we have

$$(d_h^{2,k+1})^2 \leq \rho(d_h^{2,k})^2 + C_{2,1}\eta + C_{2,2}\rho^k + \frac{C_{2,3}}{N^{\frac{2}{3}}},$$

where $C_{2,1}, C_{2,2}$, and $C_{2,3}$ are all constants. Hence we have

$$(d_h^{2,k+1})^2 - \frac{C_{2,1}\eta + C_{2,2}\rho^k + \frac{C_{2,3}}{N^{\frac{2}{3}}}}{1-\rho} \leq \rho((d_h^{2,k})^2 - \frac{C_{2,1}\eta + C_{2,2}\rho^k + \frac{C_{2,3}}{N^{\frac{2}{3}}}}{1-\rho})$$
$$\leq \cdots$$
$$\leq \rho^k((d_h^{2,1})^2 - \frac{C_{2,1}\eta + C_{2,2}\rho^k + \frac{C_{2,3}}{N^{\frac{2}{3}}}}{1-\rho})$$
$$\leq \rho^k G,$$

which leads to

$$(d_h^{2,k+1})^2 \leq C'_{2,1}\eta + C'_{2,2}\rho^k + \frac{C'_{2,3}}{N^{\frac{2}{3}}}.$$

And so on, there exist constants $C'_{*,1}, C'_{*,2}$, and $C'_{*,3}$ that are independent with $\eta, k, l, N$ such that

$$(d_h^{l,k+1})^2 \leq C'_{*,1}\eta + C'_{*,2}\rho^k + \frac{C'_{*,3}}{N^{\frac{2}{3}}}, \ \forall\, l \in [L], \, k \in \mathbb{N}^*.$$

□

**Lemma 6.** *For an L-layer GNN, suppose that Assumption 1 holds. Besides, we suppose that*

1. *$(d_h^{l,1})^2$ is bounded by $G > 1$, $\forall\, l \in [L]$,*

2. *there exists $N \in \mathbb{N}^*$ such that*

$$\|\widehat{\mathbf{V}}^{l,k} - \mathbf{V}^{l,k}\|_F \leq \|\overline{\mathbf{V}}^{l,k} - \mathbf{V}^{l,k}\|_F + \frac{1}{N^{\frac{2}{3}}}, \ \forall\, l \in [L], \, k \in \mathbb{N}^*,$$

$$\|\mathbf{V}^{l,k} - \mathbf{V}^{l,k-1}\|_F \leq \frac{1}{N^{\frac{2}{3}}}, \ \forall\, k \in \mathbb{N}^*,$$

*then there exist constants $C'_{*,1}, C'_{*,2}$, and $C'_{*,3}$ that are independent with $k, l, \varepsilon^*$, and $\eta$, such that*

$$(d_v^{l,k+1})^2 \leq C'_{*,1}\eta + C'_{*,2}\rho^k + \frac{C'_{*,3}}{N^{\frac{2}{3}}}, \ \forall\, l \in [L], \, k \in \mathbb{N}^*,$$

*where $\rho = \frac{n-S}{n} < 1$, $n = |\mathcal{V}|$, and $S$ is number of sampled nodes at each iteration.*

*Proof.* Similar to the proof of Lemma 5. □

### D.4 PROOF OF THEOREM 2: APPROXIMATION ERRORS OF MINI-BATCH GRADIENTS

In this subsection, we focus on the mini-batch gradients computed by LMC, i.e.,

$$\widetilde{\mathbf{g}}_w(w^k; \mathcal{V}_{\mathcal{B}}^k) = \frac{1}{|\mathcal{V}_L^k|} \sum_{v_j \in \mathcal{V}_L^k} \nabla_w \ell_{w^k}(\overline{\mathbf{h}}_j^k, y_j)$$

and

$$\widetilde{\mathbf{g}}_{\theta^l}(\theta^{l,k}; \mathcal{V}_{\mathcal{B}}^k) = \frac{|\mathcal{V}|}{|\mathcal{V}_{\mathcal{B}}^k|} \sum_{v_j \in \mathcal{V}_{\mathcal{B}}^k} \left( \nabla_{\theta^l} u_{\theta^{l,k}}(\overline{\mathbf{h}}_j^{l-1,k}, \overline{\mathbf{m}}_{\mathcal{N}(v_j)}^{l-1,k}, \mathbf{x}_j) \right) \overline{\mathbf{V}}_j^{l,k}, \ l \in [L],$$

where $\mathcal{V}_{\mathcal{B}}^k$ is the sampled mini-batch and $\mathcal{V}_{L_{\mathcal{B}}}^k$ is the corresponding labeled node set at the $k$-th iteration. We denote the mini-batch gradients computed by backward SGD by

$$\mathbf{g}_w(w^k; \mathcal{V}_{\mathcal{B}}^k) = \frac{1}{|\mathcal{V}_L^k|} \sum_{v_j \in \mathcal{V}_L^k} \nabla_w \ell_{w^k}(\mathbf{h}_j^k, y_j)$$

and

$$\mathbf{g}_{\theta^l}(\theta^{l,k}; \mathcal{V}_{\mathcal{B}}^k) = \frac{|\mathcal{V}|}{|\mathcal{V}_{\mathcal{B}}^k|} \sum_{v_j \in \mathcal{V}_{\mathcal{B}}^k} \left( \nabla_{\theta^l} u_{\theta^{l,k}}(\mathbf{h}_j^{l-1,k}, \mathbf{m}_{\mathcal{N}(v_j)}^{l-1,k}, \mathbf{x}_j) \right) \mathbf{V}_j^{l,k}, \ l \in [L].$$

Below, we omit the sampled subgraph $\mathcal{V}_{\mathcal{B}}^k$ and simply write the mini-batch gradients as $\widetilde{\mathbf{g}}_w(w^k)$, $\widetilde{\mathbf{g}}_{\theta^l}(\theta^{l,k})$, $\mathbf{g}_w(w^k)$, and $\mathbf{g}_{\theta^l}(\theta^{l,k})$. The approximation errors of gradients are denoted by

$$\Delta_w^k \triangleq \widetilde{\mathbf{g}}_w(w^k) - \nabla_w \mathcal{L}(w^k)$$

and

$$\Delta_{\theta^l}^k \triangleq \widetilde{\mathbf{g}}_{\theta^l}(\theta^{l,k}) - \nabla_{\theta^l} \mathcal{L}(\theta^{l,k}).$$

We restate Theorem 2 as follows.

**Theorem 4.** *For any $k \in \mathbb{N}^*$ and $l \in [L]$, the expectations of $\|\Delta_w^k\|_2^2 \triangleq \|\widetilde{\mathbf{g}}_w(w^k) - \nabla_w \mathcal{L}(w^k)\|_2^2$ and $\|\Delta_{\theta^l}^k\|_2^2 \triangleq \|\widetilde{\mathbf{g}}_{\theta^l}(\theta^{l,k}) - \nabla_{\theta^l} \mathcal{L}(\theta^{l,k})\|_2^2$ have the bias-variance decomposition*

$$\mathbb{E}[\|\Delta_w^k\|_2^2] = (\mathrm{Bias}(\widetilde{\mathbf{g}}_w(w^k)))^2 + \mathrm{Var}(\mathbf{g}_w(w^k)),$$
$$\mathbb{E}[\|\Delta_{\theta^l}^k\|_2^2] = (\mathrm{Bias}(\widetilde{\mathbf{g}}_{\theta^l}(\theta^{l,k})))^2 + \mathrm{Var}(\mathbf{g}_{\theta^l}(\theta^{l,k})),$$

*where*

$$\mathrm{Bias}(\widetilde{\mathbf{g}}_w(w^k)) = \left( \mathbb{E}[\|\widetilde{\mathbf{g}}_w(w^k) - \mathbf{g}_w(w^k)\|_2^2] \right)^{\frac{1}{2}},$$
$$\mathrm{Var}(\mathbf{g}_w(w^k)) = \mathbb{E}[\|\mathbf{g}_w(w^k) - \nabla_w \mathcal{L}(w^k)\|_2^2],$$
$$\mathrm{Bias}(\widetilde{\mathbf{g}}_{\theta^l}(\theta^{l,k})) = \left( \mathbb{E}[\|\widetilde{\mathbf{g}}_{\theta^l}(\theta^{l,k}) - \mathbf{g}_{\theta^l}(\theta^{l,k})\|_2^2] \right)^{\frac{1}{2}},$$
$$\mathrm{Var}(\mathbf{g}_{\theta^l}(\theta^{l,k})) = \mathbb{E}[\|\mathbf{g}_{\theta^l}(\theta^{l,k}) - \nabla_{\theta^l} \mathcal{L}(\theta^{l,k})\|_2^2].$$

*Suppose that Assumption 1 holds, then with $\eta = \mathcal{O}(\varepsilon^2)$ and $\beta_i = \mathcal{O}(\varepsilon^2)$, $i \in [n]$, there exist $C > 0$ and $\rho \in (0, 1)$ such that for any $k \in \mathbb{N}^*$ and $l \in [L]$, the bias terms can be bounded as*

$$\mathrm{Bias}(\widetilde{\mathbf{g}}_w(w^k)) \le C\varepsilon + C\rho^{\frac{k-1}{2}},$$
$$\mathrm{Bias}(\widetilde{\mathbf{g}}_{\theta^l}(\theta^{l,k})) \le C\varepsilon + C\rho^{\frac{k-1}{2}}.$$

*Hence we have*

$$\mathbb{E}[\|\Delta_w^k\|_2] \le C\varepsilon + C\rho^{\frac{k-1}{2}} + \mathrm{Var}(\mathbf{g}_w(w^k))^{\frac{1}{2}},$$
$$\mathbb{E}[\|\Delta_{\theta^l}^k\|_2] \le C\varepsilon + C\rho^{\frac{k-1}{2}} + \mathrm{Var}(\mathbf{g}_{\theta^l}(\theta^{l,k}))^{\frac{1}{2}}.$$

**Lemma 7.** *Suppose that Assumption 1 holds. For any $k \in \mathbb{N}^*$, the difference between $\widetilde{\mathbf{g}}_w(w^k)$ and $\mathbf{g}_w(w^k)$ can be bounded as*

$$\|\widetilde{\mathbf{g}}_w(w^k) - \mathbf{g}_w(w^k)\|_2 \le \gamma \|\overline{\mathbf{H}}^{L,k} - \mathbf{H}^{L,k}\|_F.$$

*Proof.* We have

$$\|\widetilde{\mathbf{g}}_w(w^k) - \mathbf{g}_w(w^k)\|_2 = \frac{1}{|\mathcal{V}_L^k|} \| \sum_{v_j \in \mathcal{V}_L^k} \nabla_w \ell_{w^k}(\overline{\mathbf{h}}_j^{L,k}, y_j) - \nabla_w \ell_{w^k}(\mathbf{h}_j^{L,k}, y_j)\|_2$$

$$\leq \frac{1}{|\mathcal{V}_L^k|} \sum_{v_j \in \mathcal{V}_L^k} \|\nabla_w \ell_{w^k}(\overline{\mathbf{h}}_j^{L,k}, y_j) - \nabla_w \ell_{w^k}(\mathbf{h}_j^{L,k}, y_j)\|_2$$

$$\leq \frac{\gamma}{|\mathcal{V}_L^k|} \sum_{v_j \in \mathcal{V}_L^k} \|\overline{\mathbf{h}}_j^{L,k} - \mathbf{h}_j^{L,k}\|_2$$

$$\leq \frac{\gamma}{|\mathcal{V}_L^k|} \sum_{v_j \in \mathcal{V}_L^k} \|\overline{\mathbf{H}}^{L,k} - \mathbf{H}^{L,k}\|_F$$

$$= \frac{\gamma}{|\mathcal{V}_L^k|} \cdot |\mathcal{V}_L^k| \cdot \|\overline{\mathbf{H}}^{L,k} - \mathbf{H}^{L,k}\|_F$$

$$= \gamma \|\overline{\mathbf{H}}^{L,k} - \mathbf{H}^{L,k}\|_F$$

□

**Lemma 8.** *Suppose that Assumption 1 holds. For any $k \in \mathbb{N}^*$ and $l \in [L]$, the difference between $\widetilde{\mathbf{g}}_{\theta^l}(\theta^{l,k})$ and $\mathbf{g}_{\theta^l}(\theta^{l,k})$ can be bounded as*

$$\|\widetilde{\mathbf{g}}_{\theta^l}(\theta^{l,k}) - \mathbf{g}_{\theta^l}(\theta^{l,k})\|_2 \leq |\mathcal{V}|G\|\overline{\mathbf{V}}^{l,k} - \mathbf{V}^{l,k}\|_F + |\mathcal{V}|G\gamma\|\overline{\mathbf{H}}^{l,k} - \mathbf{H}^{l,k}\|_F.$$

*Proof.* As $\|\mathbf{Aa}-\mathbf{Bb}\|_2 \leq \|\mathbf{A}\|_F\|\mathbf{a}-\mathbf{b}\|_2+\|\mathbf{A}-\mathbf{B}\|_F\|\mathbf{b}\|_2$, we can bound $\|\widetilde{\mathbf{g}}_{\theta^l}(\theta^{l,k})-\mathbf{g}_{\theta^l}(\theta^{l,k})\|_2$ by

$$\|\widetilde{\mathbf{g}}_{\theta^l}(\theta^{l,k}) - \mathbf{g}_{\theta^l}(\theta^{l,k})\|_2$$

$$\leq \frac{|\mathcal{V}|}{|\mathcal{V}_\mathcal{B}^k|} \sum_{v_i \in \mathcal{V}_\mathcal{B}^k} \| \left(\nabla_{\theta^l} u_{\theta^{l,k}}(\overline{\mathbf{h}}_j^{l-1,k}, \overline{\mathbf{m}}_{\mathcal{N}(v_j)}^{l-1,k}, \mathbf{x}_j)\right) \overline{\mathbf{V}}_j^{l,k} - \left(\nabla_{\theta^l} u_{\theta^{l,k}}(\mathbf{h}_j^{l-1,k}, \mathbf{m}_{\mathcal{N}(v_j)}^{l-1,k}, \mathbf{x}_j)\right) \mathbf{V}_j^{l,k}\|_2$$

$$\leq |\mathcal{V}| \max_{v_i \in \mathcal{V}_\mathcal{B}^k} \| \left(\nabla_{\theta^l} u_{\theta^{l,k}}(\overline{\mathbf{h}}_j^{l-1,k}, \overline{\mathbf{m}}_{\mathcal{N}(v_j)}^{l-1,k}, \mathbf{x}_j)\right) \overline{\mathbf{V}}_j^{l,k} - \left(\nabla_{\theta^l} u_{\theta^{l,k}}(\mathbf{h}_j^{l-1,k}, \mathbf{m}_{\mathcal{N}(v_j)}^{l-1,k}, \mathbf{x}_j)\right) \mathbf{V}_j^{l,k}\|_2$$

$$\leq |\mathcal{V}| \max_{v_i \in \mathcal{V}_\mathcal{B}^k} \{\|\nabla_{\theta^l} u_{\theta^{l,k}}(\overline{\mathbf{h}}_j^{l-1,k}, \overline{\mathbf{m}}_{\mathcal{N}(v_j)}^{l-1,k}, \mathbf{x}_j)\|_F\|\overline{\mathbf{V}}_j^{l,k} - \mathbf{V}_j^{l,k}\|_2$$

$$+ \|\nabla_{\theta^l} u_{\theta^{l,k}}(\overline{\mathbf{h}}_j^{l-1,k}, \overline{\mathbf{m}}_{\mathcal{N}(v_j)}^{l-1,k}, \mathbf{x}_j) - \nabla_{\theta^l} u_{\theta^{l,k}}(\mathbf{h}_j^{l-1,k}, \mathbf{m}_{\mathcal{N}(v_j)}^{l-1,k}, \mathbf{x}_j)\|_F\|\mathbf{V}_j^{l,k}\|_2\}$$

$$\leq |\mathcal{V}|G\|\overline{\mathbf{V}}^{l,k} - \mathbf{V}^{l,k}\|_F + |\mathcal{V}|G\gamma\|\overline{\mathbf{H}}^{l,k} - \mathbf{H}^{l,k}\|_F.$$

□

**Lemma 9.** *For an $L$-layer ConvGNN, suppose that Assumption 1 holds. For any $N \in \mathbb{N}^*$, by letting*

$$\eta \leq \frac{1}{(2\gamma)^L G} \frac{1}{N^{\frac{2}{3}}} = \mathcal{O}(\frac{1}{N^{\frac{2}{3}}})$$

*and*

$$\beta_i \leq \frac{1}{2G} \frac{1}{N^{\frac{2}{3}}} = \mathcal{O}(\frac{1}{N^{\frac{2}{3}}}), \ i \in [n],$$

*there exists $G_{2,*} > 0$ and $\rho \in (0,1)$ such that for any $k \in \mathbb{N}^*$ we have*

$$\mathbb{E}[\|\Delta_w^k\|_2^2] = (\text{Bias}(\widetilde{\mathbf{g}}_w(w^k)))^2 + \text{Var}(\mathbf{g}_w(w^k)),$$

$$\mathbb{E}[\|\Delta_{\theta^l}^k\|_2^2] = (\text{Bias}(\widetilde{\mathbf{g}}_{\theta^l}(\theta^{l,k})))^2 + \text{Var}(\mathbf{g}_{\theta^l}(\theta^{l,k})),$$

*where*

$$\text{Var}(\mathbf{g}_w(w^k)) = \mathbb{E}[\|\mathbf{g}_w(w^k) - \nabla_w \mathcal{L}(w^k)\|_2^2],$$

$$\text{Bias}(\widetilde{\mathbf{g}}_w(w^k)) = \left(\mathbb{E}[\|\widetilde{\mathbf{g}}_w(w^k) - \mathbf{g}_w(w^k)\|_2^2]\right)^{\frac{1}{2}},$$

$$\text{Var}(\mathbf{g}_{\theta^l}(\theta^{l,k})) = \mathbb{E}[\|\mathbf{g}_{\theta^l}(\theta^{l,k}) - \nabla_{\theta^l} \mathcal{L}(\theta^{l,k})\|_2^2],$$

$$\text{Bias}(\widetilde{\mathbf{g}}_{\theta^l}(\theta^{l,k})) = \left(\mathbb{E}[\|\widetilde{\mathbf{g}}_{\theta^l}(\theta^{l,k}) - \mathbf{g}_{\theta^l} \mathcal{L}(\theta^{l,k})\|_2^2]\right)^{\frac{1}{2}}$$

*and*

$$\text{Bias}(\widetilde{\mathbf{g}}_w(w^k)) \leq G_{2,*}(\eta^{\frac{1}{2}} + \rho^{\frac{k-1}{2}} + \frac{1}{N^{\frac{1}{3}}}),$$

$$\text{Bias}(\widetilde{\mathbf{g}}_{\theta^l}(\theta^{l,k})) \leq G_{2,*}(\eta^{\frac{1}{2}} + \rho^{\frac{k-1}{2}} + \frac{1}{N^{\frac{1}{3}}}).$$

*Proof.* By Lemmas 1 and 2 we know that

$$\|\mathbf{H}^{l,k+1} - \mathbf{H}^{l,k}\|_F < \frac{1}{N^{\frac{2}{3}}}, \ \forall l \in [L], \ k \in \mathbb{N}^*,$$

$$\|\mathbf{V}^{l,k+1} - \mathbf{V}^{l,k}\|_F < \frac{1}{N^{\frac{2}{3}}}, \ \forall l \in [L], \ k \in \mathbb{N}^*.$$

By Lemmas 3 and 4 we know that for any $k \in \mathbb{N}^*$ and $l \in [L]$ we have

$$\|\widehat{\mathbf{H}}^{l,k} - \mathbf{H}^{l,k}\|_F \leq \|\overline{\mathbf{H}}^{l,k} - \mathbf{H}^{l,k}\|_F + \frac{1}{N^{\frac{2}{3}}}$$

and

$$\|\widehat{\mathbf{V}}^{l,k} - \mathbf{V}^{l,k}\|_F \leq \|\overline{\mathbf{V}}^{l,k} - \mathbf{V}^{l,k}\|_F + \frac{1}{N^{\frac{2}{3}}}.$$

Thus, by Lemmas 5 and 6 we know that there exist $C'_{*,1}$, $C'_{*,2}$, and $C'_{*,3}$ that do not depend on $k, l, \eta, N$ such that for $\forall l \in [L]$ and $k \in \mathbb{N}^*$ hold

$$d_h^{l,k} \leq \sqrt{C'_{*,1}\eta + C'_{*,2}\rho^{k-1} + \frac{C'_{*,3}}{N^{\frac{2}{3}}}}$$

$$\leq \sqrt{C'_{*,1}}\eta^{\frac{1}{2}} + \sqrt{C'_{*,2}}\rho^{\frac{k-1}{2}} + \sqrt{C'_{*,3}}\frac{1}{N^{\frac{1}{3}}}$$

and

$$d_v^{l,k} \leq \sqrt{C'_{*,1}\eta + C'_{*,2}\rho^{k-1} + \frac{C'_{*,3}}{N^{\frac{2}{3}}}}$$

$$\leq \sqrt{C'_{*,1}}\eta^{\frac{1}{2}} + \sqrt{C'_{*,2}}\rho^{\frac{k-1}{2}} + \sqrt{C'_{*,3}}\frac{1}{N^{\frac{1}{3}}}.$$

We can decompose $\|\Delta_w^k\|_2^2$ as

$$\|\Delta_w^k\|_2^2$$
$$= \|\widetilde{\mathbf{g}}_w(w^k) - \nabla_w\mathcal{L}(w^k)\|_2^2$$
$$= \|\widetilde{\mathbf{g}}_w(w^k) - \mathbf{g}_w(w^k) + \mathbf{g}_w(w^k) - \nabla_w\mathcal{L}(w^k)\|_2^2$$
$$= \|\widetilde{\mathbf{g}}_w(w^k) - \mathbf{g}_w(w^k)\|_2^2 + \|\mathbf{g}_w(w^k) - \nabla_w\mathcal{L}(w^k)\|_2^2$$
$$+ 2\langle\|\widetilde{\mathbf{g}}_w(w^k) - \mathbf{g}_w(w^k), \mathbf{g}_w(w^k) - \nabla_w\mathcal{L}(w^k)\rangle.$$

We take expectation of both sides of the above expression, leading to

$$\mathbb{E}[\|\Delta_w^k\|_2^2] = (\text{Bias}(\widetilde{\mathbf{g}}_w(w^k)))^2 + \text{Var}(\mathbf{g}_w(w^k)), \tag{16}$$

where

$$\text{Bias}(\widetilde{\mathbf{g}}_w(w^k)) = \left(\mathbb{E}[\|\widetilde{\mathbf{g}}_w(w^k) - \mathbf{g}_w(w^k)\|_2^2]\right)^{\frac{1}{2}},$$

$$\text{Var}(\mathbf{g}_w(w^k)) = \mathbb{E}[\|\mathbf{g}_w(w^k) - \nabla_w\mathcal{L}(w^k)\|_2^2]$$

as

$$\mathbb{E}[\langle\widetilde{\mathbf{g}}_w(w^k) - \mathbf{g}_w(w^k), \mathbf{g}_w(w^k) - \nabla_w\mathcal{L}(w^k)\rangle] = 0.$$

By Lemma 7, we can bound the bias term as

$$
\begin{aligned}
\mathrm{Bias}(\widetilde{\mathbf{g}}_w(w^k)) &= \left(\mathbb{E}[\|\widetilde{\mathbf{g}}_w(w^k) - \mathbf{g}_w(w^k)\|_2^2]\right)^{\frac{1}{2}} \\
&\leq \left(\gamma^2 \mathbb{E}[\|\overline{\mathbf{H}}^{L,k} - \mathbf{H}^{L,k}\|_F^2]\right)^{\frac{1}{2}} \\
&= \gamma \cdot d_h^{L,k} \\
&\leq \gamma(\sqrt{C'_{*,1}}\eta^{\frac{1}{2}} + \sqrt{C'_{*,2}}\rho^{\frac{k-1}{2}} + \sqrt{C'_{*,3}}\frac{1}{N^{\frac{1}{3}}}) \\
&\leq G_{2,1}(\eta^{\frac{1}{2}} + \rho^{\frac{k-1}{2}} + \frac{1}{N^{\frac{1}{3}}}),
\end{aligned}
\tag{17}
$$

where $G_{2,1} = \gamma \max\{\sqrt{C'_{*,1}}, \sqrt{C'_{*,2}}, \sqrt{C'_{*,3}}\}$.

Similar to Eq. equation 16, we can decompose $\mathbb{E}[\|\Delta_{\theta^l}^k\|_2^2]$ as

$$
\mathbb{E}[\|\Delta_{\theta^l}^k\|_2^2] = (\mathrm{Bias}(\widetilde{\mathbf{g}}_{\theta^l}(\theta^{l,k})))^2 + \mathrm{Var}(\mathbf{g}_{\theta^l}(\theta^{l,k})),
$$

where

$$
\begin{aligned}
\mathrm{Bias}(\widetilde{\mathbf{g}}_{\theta^l}(\theta^{l,k})) &= \left(\mathbb{E}[\|\widetilde{\mathbf{g}}_{\theta^l}(\theta^{l,k}) - \mathbf{g}_{\theta^l}(\theta^{l,k})\|_2^2]\right)^{\frac{1}{2}}, \\
\mathrm{Var}(\mathbf{g}_{\theta^l}(\theta^{l,k})) &= \mathbb{E}[\|\mathbf{g}_{\theta^l}(\theta^{l,k}) - \nabla_{\theta^l}(\theta^{l,k}))\|_2^2].
\end{aligned}
$$

By Lemma 8, we can bound the bias term as

$$
\begin{aligned}
\mathrm{Bias}(\widetilde{\mathbf{g}}_{\theta^l}(\theta^{l,k})) &= \left(\mathbb{E}[\|\widetilde{\mathbf{g}}_{\theta^l}(\theta^{l,k}) - \mathbf{g}_{\theta^l}(\theta^{l,k})\|_2^2]\right)^{\frac{1}{2}} \\
&\leq (2|\mathcal{V}|^2 G^2 \mathbb{E}[\|\overline{\mathbf{V}}^{l,k} - \mathbf{V}^{l,k}\|_F^2] \\
&\quad + 2|\mathcal{V}|^2 G^2 \gamma^2 \mathbb{E}[\|\overline{\mathbf{H}}^{l,k} - \mathbf{H}^{l,k}\|_F^2])^{\frac{1}{2}} \\
&\leq \sqrt{2}|\mathcal{V}|G d_v^{l,k} + \sqrt{2}|\mathcal{V}|G\gamma d_h^{l,k} \\
&\leq G_{2,2}(\eta^{\frac{1}{2}} + \rho^{\frac{k-1}{2}} + \frac{1}{N^{\frac{1}{3}}}),
\end{aligned}
\tag{18}
$$

where $G_{2,2} = \sqrt{2}|\mathcal{V}|G(1+\gamma)\max\{\sqrt{C'_{*,1}}, \sqrt{C'_{*,2}}, \sqrt{C'_{*,3}}\}$.

Let $G_{2,*} = \max\{G_{2,1}, G_{2,2}\}$, then we have

$$
\begin{aligned}
\mathrm{Bias}(\widetilde{\mathbf{g}}_w(w^k)) &\leq G_{2,*}(\eta^{\frac{1}{2}} + \rho^{\frac{k-1}{2}} + \frac{1}{N^{\frac{1}{3}}}), \\
\mathrm{Bias}(\widetilde{\mathbf{g}}_{\theta^l}(\theta^{l,k})) &\leq G_{2,*}(\eta^{\frac{1}{2}} + \rho^{\frac{k-1}{2}} + \frac{1}{N^{\frac{1}{3}}}).
\end{aligned}
$$

$\square$

By letting $\varepsilon = \frac{1}{N^{\frac{1}{3}}}$ and $C = 2G_{2,*}$, we have

$$
\begin{aligned}
\mathrm{Bias}(\widetilde{\mathbf{g}}_w(w^k)) &\leq C\varepsilon + C\rho^{\frac{k-1}{2}}, \\
\mathrm{Bias}(\widetilde{\mathbf{g}}_{\theta^l}(\theta^{l,k})) &\leq C\varepsilon + C\rho^{\frac{k-1}{2}},
\end{aligned}
$$

which leads to

$$
\begin{aligned}
\mathbb{E}[\|\Delta_w^k\|_2] &\leq \left(\mathbb{E}[\|\Delta_w^k\|_2^2]\right)^{\frac{1}{2}} \\
&\leq \left((\mathrm{Bias}(\widetilde{\mathbf{g}}_w(w^k)))^2 + \mathrm{Var}(\mathbf{g}_w(w^k))\right)^{\frac{1}{2}} \\
&\leq \mathrm{Bias}(\widetilde{\mathbf{g}}_w(w^k)) + \mathrm{Var}(\mathbf{g}_w(w^k))^{\frac{1}{2}} \\
&\leq C\varepsilon + C\rho^{\frac{k-1}{2}} + \mathrm{Var}(\mathbf{g}_w(w^k))^{\frac{1}{2}}
\end{aligned}
$$

and

$$
\begin{aligned}
\mathbb{E}[\|\Delta_{\theta^l}^k\|_2] &\leq \left(\mathbb{E}[\|\Delta_{\theta^l}^k\|_2^2]\right)^{\frac{1}{2}} \\
&\leq \left((\operatorname{Bias}(\widetilde{\mathbf{g}}_{\theta^l}(\theta^{l,k})))^2 + \operatorname{Var}(\mathbf{g}_{\theta^l}(\theta^{l,k}))\right)^{\frac{1}{2}} \\
&\leq \operatorname{Bias}(\widetilde{\mathbf{g}}_{\theta^l}(\theta^{l,k})) + \operatorname{Var}(\mathbf{g}_{\theta^l}(\theta^{l,k}))^{\frac{1}{2}} \\
&\leq C\varepsilon + C\rho^{\frac{k-1}{2}} + \operatorname{Var}(\mathbf{g}_{\theta^l}(\theta^{l,k}))^{\frac{1}{2}}.
\end{aligned}
$$

Theorem 2 and Theorem 4 follow immediately.

### D.5 PROOF OF THEOREM 3: CONVERGENCE GUARANTEES

In this subsection, we give the convergence guarantees of LMC. We first give sufficient conditions for convergence.

**Lemma 10.** *Suppose that function $f : \mathbb{R}^n \to \mathbb{R}$ is continuously differentiable. Consider an optimization algorithm with any bounded initialization $\mathbf{x}^1$ and an update rule in the form of*

$$
\mathbf{x}^{k+1} = \mathbf{x}^k - \eta \mathbf{d}(\mathbf{x}^k),
$$

*where $\eta > 0$ is the learning rate and $\mathbf{d}(\mathbf{x}^k)$ is the estimated gradient that can be seen as a stochastic vector depending on $\mathbf{x}^k$. Let the estimation error of the gradient be $\Delta^k = \mathbf{d}(\mathbf{x}^k) - \nabla f(\mathbf{x}^k)$. Suppose that*

1. *the optimal value $f^* = \inf_{\mathbf{x}} f(\mathbf{x})$ is bounded;*

2. *the gradient of $f$ is $\gamma$-Lipschitz, i.e.,*

$$
\|\nabla f(\mathbf{y}) - \nabla f(\mathbf{x})\|_2 \leq \gamma \|\mathbf{y} - \mathbf{x}\|_2, \ \forall \mathbf{x}, \mathbf{y} \in \mathbb{R}^n;
$$

3. *there exists $G_0 > 0$ that does not depend on $\eta$ such that*

$$
\mathbb{E}[\|\Delta^k\|_2^2] \leq G_0, \ \forall k \in \mathbb{N}^*;
$$

4. *there exists $N \in \mathbb{N}^*$ and $\rho \in (0, 1)$ that do not depend on $\eta$ such that*

$$
|\mathbb{E}[\langle \nabla f(\mathbf{x}^k), \Delta^k \rangle]| \leq G_0(\eta^{\frac{1}{2}} + \rho^{\frac{k-1}{2}} + \frac{1}{N^{\frac{1}{3}}}), \ \forall k \in \mathbb{N}^*,
$$

   *where $G_0$ is the same constant as that in Condition 3,*

*then by letting $\eta = \min\{\frac{1}{\gamma}, \frac{1}{N^{\frac{2}{3}}}\}$, we have*

$$
\mathbb{E}[\|\nabla f(\mathbf{x}^R)\|_2^2] \leq \frac{2(f(\mathbf{x}^1) - f^* + G_0)}{N^{\frac{1}{3}}} + \frac{\gamma G_0}{N^{\frac{2}{3}}} + \frac{G_0}{N(1 - \sqrt{\rho})} = \mathcal{O}(\frac{1}{N^{\frac{1}{3}}}),
$$

*where $R$ is chosen uniformly from $[N]$.*

*Proof.* As the gradient of $f$ is $\gamma$-Lipschitz, we have

$$
\begin{aligned}
f(\mathbf{y}) &= f(\mathbf{x}) + \int_{\mathbf{x}}^{\mathbf{y}} \nabla f(\mathbf{z}) \, d\mathbf{z} \\
&= f(\mathbf{x}) + \int_0^1 \langle \nabla f(\mathbf{x} + t(\mathbf{y} - \mathbf{x})), \mathbf{y} - \mathbf{x} \rangle \, dt \\
&= f(\mathbf{x}) + \langle \nabla f(\mathbf{x}), \mathbf{y} - \mathbf{x} \rangle + \int_0^1 \langle \nabla f(\mathbf{x} + t(\mathbf{y} - \mathbf{x})) - \nabla f(\mathbf{x}), \mathbf{y} - \mathbf{x} \rangle \, dt \\
&\leq f(\mathbf{x}) + \langle \nabla f(\mathbf{x}), \mathbf{y} - \mathbf{x} \rangle + \int_0^1 \|\nabla f(\mathbf{x} + t(\mathbf{y} - \mathbf{x})) - \nabla f(\mathbf{x})\|_2 \|\mathbf{y} - \mathbf{x}\|_2 \, dt \\
&\leq f(\mathbf{x}) + \langle \nabla f(\mathbf{x}), \mathbf{y} - \mathbf{x} \rangle + \int_0^1 \gamma t \|\mathbf{y} - \mathbf{x}\|_2^2 \, dt \\
&\leq f(\mathbf{x}) + \langle \nabla f(\mathbf{x}), \mathbf{y} - \mathbf{x} \rangle + \frac{\gamma}{2} \|\mathbf{y} - \mathbf{x}\|_2^2,
\end{aligned}
$$

Then, we have

$$f(\mathbf{x}^{k+1})$$

$$\leq f(\mathbf{x}^k) + \langle \nabla f(\mathbf{x}^k), \mathbf{x}^{k+1} - \mathbf{x}^k \rangle + \frac{\gamma}{2}\|\mathbf{x}^{k+1} - \mathbf{x}^k\|_2^2$$

$$= f(\mathbf{x}^k) - \eta \langle \nabla f(\mathbf{x}^k), \mathbf{d}(\mathbf{x}^k) \rangle + \frac{\eta^2 \gamma}{2}\|\mathbf{d}(\mathbf{x}^k)\|_2^2$$

$$= f(\mathbf{x}^k) - \eta \langle \nabla f(\mathbf{x}^k), \Delta^k \rangle - \eta \|\nabla f(\mathbf{x}^k)\|_2^2 + \frac{\eta^2 \gamma}{2}(\|\Delta^k\|_2^2 + \|\nabla f(\mathbf{x}^k)\|_2^2 + 2\langle \Delta^k, \nabla f(\mathbf{x}^k) \rangle)$$

$$= f(\mathbf{x}^k) - \eta(1 - \eta\gamma)\langle \nabla f(\mathbf{x}^k), \Delta^k \rangle - \eta(1 - \frac{\eta\gamma}{2})\|\nabla f(\mathbf{x}^k)\|_2^2 + \frac{\eta^2 \gamma}{2}\|\Delta^k\|_2^2.$$

By taking expectation of both sides, we have

$$\mathbb{E}[f(\mathbf{x}^{k+1})]$$

$$\leq \mathbb{E}[f(\mathbf{x}^k)] - \eta(1 - \eta\gamma)\mathbb{E}[\langle \nabla f(\mathbf{x}^k), \Delta^k \rangle] - \eta(1 - \frac{\eta\gamma}{2})\mathbb{E}[\|\nabla f(\mathbf{x}^k)\|_2^2] + \frac{\eta^2 \gamma}{2}\mathbb{E}[\|\Delta^k\|_2^2].$$

By summing up the above inequalities for $k \in [N]$ and dividing both sides by $N\eta(1 - \frac{\eta\gamma}{2})$, we have

$$\frac{\sum_{k=1}^N \mathbb{E}[\|\nabla f(\mathbf{x}^k)\|_2^2]}{N}$$

$$\leq \frac{f(\mathbf{x}^1) - \mathbb{E}[f(\mathbf{x}^N)]}{N\eta(1 - \frac{\eta\gamma}{2})} + \frac{\eta\gamma}{2 - \eta\gamma}\frac{\sum_{k=1}^N \mathbb{E}[\|\Delta^k\|_2^2]}{N} - \frac{(1 - \eta\gamma)}{(1 - \frac{\eta\gamma}{2})}\frac{\sum_{k=1}^N \mathbb{E}[\langle \nabla f(\mathbf{x}^k), \Delta^k \rangle]}{N}$$

$$\leq \frac{f(\mathbf{x}^1) - f^*}{N\eta(1 - \frac{\eta\gamma}{2})} + \frac{\eta\gamma}{2 - \eta\gamma}\frac{\sum_{k=1}^N \mathbb{E}[\|\Delta^k\|_2^2]}{N} + \frac{\sum_{k=1}^N |\mathbb{E}[\langle \nabla f(\mathbf{x}^k), \Delta^k \rangle]|}{N},$$

where the second inequality comes from $\eta\gamma > 0$ and $f(\mathbf{x}^k) \geq f^*$. According to the above conditions, we have

$$\frac{\sum_{k=1}^N \mathbb{E}[\|\nabla f(\mathbf{x}^k)\|_2^2]}{N} \leq \frac{f(\mathbf{x}^1) - f^*}{N\eta(1 - \frac{\eta\gamma}{2})} + \frac{\eta\gamma}{2 - \eta\gamma}G_0 + G_0\sum_{k=1}^N \frac{\eta^{\frac{1}{2}} + \rho^{\frac{k-1}{2}}}{N} + \frac{G_0}{N^{\frac{1}{3}}}$$

$$\leq \frac{f(\mathbf{x}^1) - f^*}{N\eta(1 - \frac{\eta\gamma}{2})} + \frac{\eta\gamma}{2 - \eta\gamma}G_0 + \eta^{\frac{1}{2}}G_0 + \frac{G_0}{N}\sum_{k=1}^\infty \rho^{\frac{k-1}{2}} + \frac{G_0}{N^{\frac{1}{3}}}$$

$$= \frac{f(\mathbf{x}^1) - f^*}{N\eta(1 - \frac{\eta\gamma}{2})} + \frac{\eta\gamma}{2 - \eta\gamma}G_0 + \eta^{\frac{1}{2}}G_0 + \frac{G_0}{N(1 - \sqrt{\rho})} + \frac{G_0}{N^{\frac{1}{3}}}.$$

Notice that

$$\mathbb{E}[\|\nabla f(\mathbf{x}^R)\|_2^2] = \mathbb{E}_R[\mathbb{E}[\|\nabla f(\mathbf{x}^R)\|_2^2 \mid R]] = \frac{\sum_{k=1}^N \mathbb{E}[\|\nabla f(\mathbf{x}^k)\|_2^2]}{N},$$

where $R$ is uniformly chosen from $[N]$, hence we have

$$\mathbb{E}[\|\nabla f(\mathbf{x}^R)\|_2^2] \leq \frac{f(\mathbf{x}^1) - f^*}{N\eta(1 - \frac{\eta\gamma}{2})} + \frac{\eta\gamma}{2 - \eta\gamma}G_0 + \eta^{\frac{1}{2}}G_0 + \frac{G_0}{N(1 - \sqrt{\rho})} + \frac{G_0}{N^{\frac{1}{3}}}.$$

By letting $\eta = \min\{\frac{1}{\gamma}, \frac{1}{N^{\frac{2}{3}}}\}$, we have

$$\mathbb{E}[\|\nabla f(\mathbf{x}^R)\|_2^2] \leq \frac{2(f(\mathbf{x}^1) - f^*)}{N^{\frac{1}{3}}} + \frac{\gamma G_0}{N^{\frac{2}{3}}} + \frac{G_0}{N^{\frac{1}{3}}} + \frac{G_0}{N(1 - \sqrt{\rho})} + \frac{G_0}{N^{\frac{1}{3}}}$$

$$\leq \frac{2(f(\mathbf{x}^1) - f^* + G_0)}{N^{\frac{1}{3}}} + \frac{\gamma G_0}{N^{\frac{2}{3}}} + \frac{G_0}{N(1 - \sqrt{\rho})}$$

$$= \mathcal{O}(\frac{1}{N^{\frac{1}{3}}}).$$

$$\square$$

Given an $L$-layer GNN, forllowing (Chen et al., 2018a), we directly assume that:

1. the optimal value

$$\mathcal{L}^* = \inf_{w,\theta^1,\ldots,\theta^L} \mathcal{L}$$

is bounded by $G > 1$;

2. the gradients of $\mathcal{L}$ with respect to parameters $w$ and $\theta^l$, i.e.,

$$\nabla_w \mathcal{L}, \nabla_{\theta^l} \mathcal{L}$$

are $\gamma$-Lipschitz for $\forall l \in [L]$.

To show the convergence of LMC by Lemma 10, it suffices to show that

3. there exists $G_1 > 0$ that does not depend on $\eta$ such that

$$\mathbb{E}[\|\Delta_w^k\|_2^2] \leq G_1, \ \forall k \in \mathbb{N}^*,$$
$$\mathbb{E}[\|\Delta_{\theta^l}^k\|_2^2] \leq G_1, \ \forall l \in [L], \ k \in \mathbb{N}^*;$$

4. for any $N \in \mathbb{N}^*$, there exist $G_2 > 0$ and $\rho \in (0,1)$ such that

$$|\mathbb{E}[\langle \nabla_w \mathcal{L}, \Delta_w^k \rangle]| \leq G_2(\eta^{\frac{1}{2}} + \rho^{\frac{k-1}{2}} + \frac{1}{N^{\frac{1}{3}}}), \ \forall k \in \mathbb{N}^*,$$
$$|\mathbb{E}[\langle \nabla_{\theta^l} \mathcal{L}, \Delta_{\theta^l}^k \rangle]| \leq G_2(\eta^{\frac{1}{2}} + \rho^{\frac{k-1}{2}} + \frac{1}{N^{\frac{1}{3}}}), \ \forall l \in [L], \ k \in \mathbb{N}^*$$

by letting

$$\eta \leq \frac{1}{(2\gamma)^L G} \frac{1}{N^{\frac{2}{3}}} = \mathcal{O}(\frac{1}{N^{\frac{2}{3}}})$$

and

$$\beta_i \leq \frac{1}{2G} \frac{1}{N^{\frac{2}{3}}} = \mathcal{O}(\frac{1}{N^{\frac{2}{3}}}), \ i \in [n].$$

**Lemma 11.** *Suppose that Assumption 1 holds, then*

$$\mathbb{E}[\|\Delta_w^k\|_2^2] \leq G_1 \triangleq 4G^2, \ \forall k \in \mathbb{N}^*,$$
$$\mathbb{E}[\|\Delta_{\theta^l}^k\|_2^2] \leq G_1 \triangleq 4G^2, \ \forall l \in [L], \ k \in \mathbb{N}^*.$$

*Proof.* We have

$$\begin{aligned} \mathbb{E}[\|\Delta_w^k\|_2^2] &= \mathbb{E}[\|\widetilde{\mathbf{g}}_w(w^k) - \nabla_w \mathcal{L}(w^k)\|_2^2] \\ &\leq 2(\mathbb{E}[\|\widetilde{\mathbf{g}}_w(w^k)\|_2^2] + \mathbb{E}[\|\nabla_w \mathcal{L}(w^k)\|_2^2]) \\ &\leq 4G^2 \end{aligned}$$

and

$$\begin{aligned} \mathbb{E}[\|\Delta_{\theta^l}^k\|_2^2] &= \mathbb{E}[\|\widetilde{\mathbf{g}}_{\theta^l}(\theta^{l,k}) - \nabla_{\theta^l} \mathcal{L}(\theta^{l,k})\|_2^2] \\ &\leq 2(\mathbb{E}[\|\widetilde{\mathbf{g}}_{\theta^l}(\theta^{l,k})\|_2^2] + \mathbb{E}[\|\nabla_{\theta^l} \mathcal{L}(\theta^{l,k})\|_2^2]) \\ &\leq 4G^2. \end{aligned}$$

$\square$

**Lemma 12.** *Suppose that Assumption 1 holds. For any $N \in \mathbb{N}^*$, there exist $G_2 > 0$ and $\rho \in (0,1)$ such that*

$$|\mathbb{E}[\langle \nabla_w \mathcal{L}, \Delta_w^k \rangle]| \leq G_2(\eta^{\frac{1}{2}} + \rho^{\frac{k-1}{2}} + \frac{1}{N^{\frac{1}{3}}}), \ \forall k \in \mathbb{N}^*,$$
$$|\mathbb{E}[\langle \nabla_{\theta^l} \mathcal{L}, \Delta_{\theta^l}^k \rangle]| \leq G_2(\eta^{\frac{1}{2}} + \rho^{\frac{k-1}{2}} + \frac{1}{N^{\frac{1}{3}}}), \ \forall l \in [L], \ k \in \mathbb{N}^*$$

*by letting*

$$\eta \leq \frac{1}{(2\gamma)^L G} \frac{1}{N^{\frac{2}{3}}} = \mathcal{O}(\frac{1}{N^{\frac{2}{3}}})$$

*and*

$$\beta_i \leq \frac{1}{2G} \frac{1}{N^{\frac{2}{3}}} = \mathcal{O}(\frac{1}{N^{\frac{2}{3}}}), \; i \in [n].$$

*Proof.* By Eqs. (17) and (18) we know that there exists $G_{2,*}$ such that for any $k \in \mathbb{N}^*$ we have

$$\mathbb{E}[\|\widetilde{\mathbf{g}}_w(w^k) - \mathbf{g}_w(w^k)\|_2] \leq \left(\mathbb{E}[\|\widetilde{\mathbf{g}}_w(w^k) - \mathbf{g}_w(w^k)\|_2^2]\right)^{\frac{1}{2}}$$
$$\leq G_{2,*}(\eta^{\frac{1}{2}} + \rho^{\frac{k-1}{2}} + \frac{1}{N^{\frac{1}{3}}})$$

and

$$\mathbb{E}[\|\widetilde{\mathbf{g}}_{\theta^l}(\theta^{l,k}) - \mathbf{g}_{\theta^l}(\theta^{l,k})\|_2] \leq \left(\mathbb{E}[\|\widetilde{\mathbf{g}}_{\theta^l}(\theta^{l,k}) - \mathbf{g}_{\theta^l}(\theta^{l,k})\|_2^2]\right)^{\frac{1}{2}}$$
$$\leq G_{2,*}(\eta^{\frac{1}{2}} + \rho^{\frac{k-1}{2}} + \frac{1}{N^{\frac{1}{3}}}),$$

where $\rho = \frac{n-S}{n} < 1$ is a constant. Hence

$$|\mathbb{E}[\langle \nabla_w \mathcal{L}, \Delta_w^k \rangle]| = |\mathbb{E}[\langle \nabla_w \mathcal{L}, \widetilde{\mathbf{g}}_w(w^k) - \nabla_w \mathcal{L}(w^k) \rangle]|$$
$$= |\mathbb{E}[\langle \nabla_w \mathcal{L}, \widetilde{\mathbf{g}}_w(w^k) - \mathbf{g}_w(w^k) \rangle]|$$
$$\leq \mathbb{E}[\|\nabla_w \mathcal{L}\|_2 \|\widetilde{\mathbf{g}}_w(w^k) - \mathbf{g}_w(w^k)\|_2]$$
$$\leq G \mathbb{E}[\|\widetilde{\mathbf{g}}_w(w^k) - \mathbf{g}_w(w^k)\|_2],$$
$$\leq G_2(\eta^{\frac{1}{2}} + \rho^{\frac{k-1}{2}} + \frac{1}{N^{\frac{1}{3}}})$$

and

$$|\mathbb{E}[\langle \nabla_{\theta^l} \mathcal{L}, \Delta_{\theta^l}^k \rangle]| = |\mathbb{E}[\langle \nabla_{\theta^l} \mathcal{L}, \widetilde{\mathbf{g}}_{\theta^l}(\theta^{l,k}) - \nabla_{\theta^l} \mathcal{L}(\theta^{l,k}) \rangle]|$$
$$= |\mathbb{E}[\langle \nabla_{\theta^l} \mathcal{L}, \widetilde{\mathbf{g}}_{\theta^l}(\theta^{l,k}) - \mathbf{g}_{\theta^l}(\theta^{l,k}) \rangle]|$$
$$\leq \mathbb{E}[\|\nabla_{\theta^l} \mathcal{L}\|_2 \|\widetilde{\mathbf{g}}_{\theta^l}(\theta^{l,k}) - \mathbf{g}_{\theta^l}(\theta^{l,k})\|_2]$$
$$\leq G \mathbb{E}[\|\widetilde{\mathbf{g}}_{\theta^l}(\theta^{l,k}) - \mathbf{g}_{\theta^l}(\theta^{l,k})\|_2]$$
$$\leq G_2(\eta^{\frac{1}{2}} + \rho^{\frac{k-1}{2}} + \frac{1}{N^{\frac{1}{3}}}),$$

where $G_2 = G G_{2,*}$. □

According to Lemmas 11 and 12, the conditions in Lemma 10 hold. By letting

$$\varepsilon = \left(\frac{2(f(\mathbf{x}^1) - f^* + G_0)}{N^{\frac{1}{3}}} + \frac{\gamma G_0}{N^{\frac{2}{3}}} + \frac{G_0}{N(1 - \sqrt{\rho})}\right)^{\frac{1}{2}} = \mathcal{O}(\frac{1}{N^{\frac{1}{6}}}),$$

Theorem 3 follows immediately.

# E MORE EXPERIMENTS

## E.1 PERFORMANCE ON SMALL DATASETS

Figure 5 reports the convergence curves GD, GAS, and LMC for GCN on three small datasets, i.e., Cora, Citeseer, and PubMed from Planetoid (Yang et al., 2016). LMC is faster than GAS, especially on the CiteSeer and PubMed datasets. Notably, the key bottleneck on the small datasets is graph sampling rather than forward and backward passes. Thus, GD is faster than GAS and LMC, as it avoids graph sampling by directly using the whole graph.

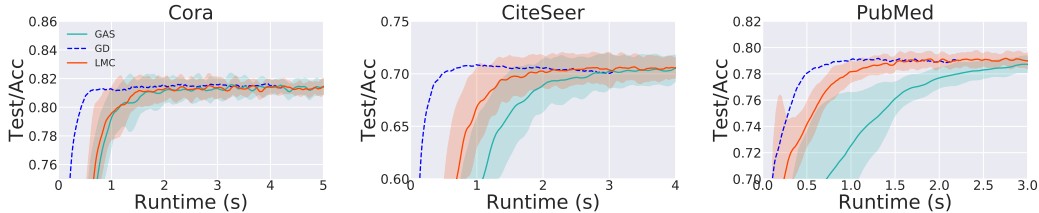

Figure 5: Testing accuracy w.r.t. runtimes (s).

## E.2    COMPARISON IN TERMS OF TRAINING TIME PER EPOCH

We evaluate the training time per epoch of CLUSTER, GAS, FM, and LMC in Table 6. Compared with GAS, LMC additionally accesses historical auxiliary variables. Inspired by GAS (Fey et al., 2021), we use the concurrent mini-batch execution to asynchronously access historical auxiliary variables. Moreover, from the convergence analysis of LMC, we can sample clusters to construct fixed subgraphs at preprocessing step (Line 2 in Algorithm 1) rather than sample clusters to construct various subgraphs at each training step[2]. This further avoids sampling costs. Finally, the training time per epoch of LMC is comparable with GAS. CLUSTER is slower than GAS and LMC, as it prunes edges in forward passes, introducing additional normalization operation for the adjacency matrix of the sampled subgraph by $[\mathbf{A}_{\mathcal{V}_{\mathcal{B}}}]_{i,j}/\sqrt{deg_{\mathcal{V}_{\mathcal{B}}}(i)deg_{\mathcal{V}_{\mathcal{B}}}(j)}$, where $deg_{\mathcal{V}_{\mathcal{B}}}(i)$ is the degree in the sampled subgraph rather than the whole graph. The normalized adjacency matrix is difficult to store and reuse, as the sampled subgraph may be different. FM is slower than other methods, as they additionally update historical embeddings in the storage for the nodes outside the mini-batches.

Table 6: Training time (s) per epoch of CLUSTER, GAS, FM, and LMC.

| Dataset & GNN | CLUSTER | GAS | FM | LMC |
|---|---|---|---|---|
| Ogbn-arxiv & GCN | 0.51 | 0.46 | 0.75 | **0.45** |
| FLICKR & GCN | 0.33 | 0.29 | 0.45 | **0.26** |
| REDDIT & GCN | 2.16 | **2.11** | 5.67 | 2.28 |
| PPI & GCN | 0.84 | **0.61** | 0.78 | 0.62 |
| Ogbn-arxiv & GCNII | — | 0.92 | 1.02 | **0.91** |
| FLICKR & GCNII | — | **1.32** | 1.44 | 1.33 |
| REDDIT & GCNII | — | **4.47** | 7.89 | 4.71 |
| PPI & GCNII | — | **4.61** | 5.38 | 4.77 |

---

[2]CLUSTER-GCN proposes to sample clusters to construct various subgraphs at each training step and LMC follows it. If a subgraph-wise sampling method prunes an edge at the current step, the GNN may observe the pruned edge at the next step by resampling subgraphs. This avoids GNN overfitting the graph which drops some important edges as shown in Section 3.2 in (Chiang et al., 2019) (we also observe that GAS achieves the accuracy of 71.5% and 71.1% under stochastic subgraph partition and fixed subgraph partition respectively on the Ogbn-arxiv dataset).

### E.3 COMPARISON IN TERMS OF MEMORY UNDER DIFFERENT BATCH SIZES

In Table 7, we report the GPU memory consumption, and the proportion of reserved messages $\sum_{k=1}^{b} \|\widetilde{\mathbf{A}}_{\mathcal{V}_{b_k}}^{alg}\|_0 / \|\widetilde{\mathbf{A}}\|_0$ in forward and backward passes for GCN, where $\widetilde{\mathbf{A}}$ is the adjacency matrix of full-batch GCN, $\widetilde{\mathbf{A}}_{\mathcal{V}_{b_k}}^{alg}$ is the adjacency matrix used in a subgraph-wise method $alg$ (e.g., CLUSTER, GAS, and LMC), and $\|\cdot\|_0$ denotes the $\ell_0$-norm. As shown in Table 7, LMC makes full use of all sampled nodes in both forward and backward passes, which is the same as full-batch GD. *Default* indicates the default batch size used in the codes and toolkits of GAS (Fey et al., 2021).

Table 7: GPU memory consumption (MB) and the proportion of reserved messages (%) in forward and backward passes of GD, CLUSTER, GAS, and LMC for training GCN. *Default* indicates the default batch size used in the codes and toolkits of GAS (Fey et al., 2021).

| Batch size | Methods | Ogbn-arxiv | FLICKR | REDDIT | PPI |
|---|---|---|---|---|---|
| Full-batch GD | | 681/**100%**/**100%** | 411/**100%**/**100%** | 2067/**100%**/**100%** | 605/**100%**/**100%** |
| 1 | CLUSTER | **177**/ 67%/ 67% | **138**/ 57%/ 57% | **428**/ 35%/ 35% | **189**/ 90%/ 90% |
| | GAS | 178/**100%**/ 67% | 168/**100%**/ 57% | 482/**100%**/ 35% | 190/**100%**/ 90% |
| | **LMC** | 207/**100%**/**100%** | 177/**100%**/**100%** | 610/**100%**/**100%** | 197/**100%**/**100%** |
| Default | CLUSTER | 424/ 83%/ 83% | 310/ 77%/ 77% | 1193/ 65%/ 65% | 212/ 91%/ 91% |
| | GAS | 452/**100%**/ 83% | 375/**100%**/ 77% | 1508/**100%**/ 65% | 214/**100%**/ 91% |
| | **LMC** | 557/**100%**/**100%** | 376/**100%**/**100%** | 1829/**100%**/**100%** | 267/**100%**/**100%** |

### E.4 ABLATION ABOUT $\beta_i$

As shown in Section A.4, $\beta_i = score(i)\alpha$ in LMC. We report the prediction performance under $\alpha \in \{0.0, 0.2, 0.4, 0.6, 0.8, 1.0\}$ and $score \in \{f(x) = x^2, f(x) = 2x - x^2, f(x) = x, f(x) = 1; x = deg_{local}(i)/deg_{global}(i)\}$ in Tables 8 and 9 respectively. When exploring the effect of a specific hyper-parameter, we fix the other hyper-parameters as their best values. Notably, $\alpha = 0$ implies that LMC directly uses the historical values as affordable without alleviating their staleness, which is the same as that in GAS. Under large batch sizes, LMC achieves the best performance with large $\beta_i = 1$, as large batch sizes improve the quality of the incomplete up-to-date messages. Under small batch sizes, LMC achieves the best performance with small $\beta_i = 0.4score_{2x-x^2}(i)$, as small learning rates alleviate the staleness of the historical values.

Table 8: Prediction performance under different $\alpha$ on the Ogbn-arxiv dataset.

| Batch Sizes | learning rates | $\alpha$ | | | | | |
|---|---|---|---|---|---|---|---|
| | | 0.0 | 0.2 | 0.4 | 0.6 | 0.8 | 1.0 |
| 1 | 1e-4 | 71.34 | 71.39 | **71.65** | 71.31 | 70.86 | 70.57 |
| 40 | 1e-2 | 69.85 | 69.12 | 69.89 | 69.61 | 69.82 | **71.44** |

Table 9: Prediction performance under different $score$ on the Ogbn-arxiv dataset.

| Batch Sizes | learning rates | $score$ | | | | |
|---|---|---|---|---|---|---|
| | | $f(x) = 2x - x^2$ | $f(x) = 1$ | $f(x) = x^2$ | $f(x) = x$ | $f(x) = sin(x)$ |
| 1 | 1e-4 | **71.35** | 70.84 | 71.32 | 71.30 | 71.13 |
| 40 | 1e-2 | 67.59 | **71.44** | 69.91 | 70.03 | 70.32 |

## F POTENTIAL SOCIETAL IMPACTS

In this paper, we propose a novel and efficient subgraph-wise sampling method for the training of GNNs, i.e., LMC. This work is promising in many practical and important scenarios such as search engine, recommendation systems, biological networks, and molecular property prediction.

Nonetheless, this work may have some potential risks. For example, using this work in search engine and recommendation systems to over-mine the behavior of users may cause undesirable privacy disclosure.

