# OpenReview forum: "LMC: Fast Training of GNNs via Subgraph Sampling with Provable Convergence"
_ICLR.cc/2023/Conference — ICLR 2023 notable top 25%_

### Official Review · Reviewer_7N9L · 2022-10-24

**Confidence:** 3
**Correctness:** 4
**Technical Novelty And Significance:** 3
**Empirical Novelty And Significance:** 3
**Recommendation:** 8

**Clarity, Quality, Novelty And Reproducibility:**

The quality of the whole research is good especially since they explain some details for their numeric experiment.

**Strength And Weaknesses:**

Strength:
1. Key idea of effective compensation is great, and they explain their idea clearly.
2. First subgraph sampling method with provable convergence for graph neural networks. Their theoretical analysis shows that mini-batch gradients are unbiased and their algorithm can converge to the stationary point of GNNs.
3. Numeric experiments are sufficient, it's worth mentioning that the experiments on compensation terms are great.

Weaknesses:
1. Some typo, denote the j-th columns of A by $A_j$ not $A_i$, forward not froward.
2. More explanations are needed for the difference of GAS and LMC, why more information keeping can get the algorithm more robust the the graph size. On some datasets, GAS performs better, why?

**Summary Of The Paper:**

This research proposes a subgraph sampling method(LMC) with provable convergence for graph neural networks. Training Graph neural networks suffer from neighbor explosion problems. Subgraph sampling methods apply to a wide range of GNN architectures by directly running GNNs on the subgraphs constructed by the sampled mini-batches and showed great empirical success in prior works, while inaccurate mini-batch gradients hurt the accuracy and performance of GNNs. The key idea of LMC is effective compensation in backward and forward passes, which computes accurate mini-batch gradients and thus accelerate convergence.

**Summary Of The Review:**

This research proposes a subgraph sampling method(LMC) with provable convergence for graph neural networks.  The key idea of LMC is effective compensation in backward and forward passes, which computes accurate mini-batch gradients and thus accelerate convergence. Numeric experiments and theoretical analysis are great, while more explanations are needed for the difference and key development with the prior works.

---

> ### Author Response · Authors · 2022-11-13
> **Response to Reviewer 7N9L**
>
> We thank the reviewer for the positive and insightful comments. We respond to each comment as follows and sincerely hope that our rebuttal could properly address your concerns. If so, we would deeply appreciate it if you could raise your score. If not, please let us know your further concerns, and we will continue actively responding to your comments and improving our submission.
>
>
> **1. Typos in the paper.**
>
> Thanks for pointing out the typos. We have corrected the typos and improve our presentation in the revised paper accordingly.
>
> **2. Why can keeping more information get the algorithm more robust to the graph size?**
>
> We do not claim that the robustness to the graph size is due to keeping more information in this paper. Table 1 in the main text and Figure 5 in Appendix E.1 in the revised version show that the LMC is applicable to both large and small graphs, which is an advantage of subgraph-wise sampling methods [1]. For your convenience, we quote Table 1 as follows.
>
> | Method & GNN |     REDDIT     |        PPI        |      FLICKR       |    Ogbn-arxiv     |
> | :----------- | :------------: | :---------------: | :---------------: | :---------------: |
> | GraphSAGE    |     95.40      |       61.20       |       50.10       |       71.49       |
> | VR-GCN       |     94.50      |       85.60      |        ---        |        ---        |
> | FastGCN      |     93.70      |        ---        |       50.40       |        ---        |
> | LADIES       |     92.80      |        ---        |        ---        |        ---        |
> | CLUSTER-GCN  |     96.60      |   *99.36*    |       48.10       |        ---        |
> | GraphSAINT   |     97.00      |       99.50       |       51.10       |        ---        |
> | SIGN         |  *96.80*  |       97.00       |       51.40       |        ---        |
> | GD & GCN     |     95.43      |       97.58       |       53.73       |       71.64       |
> | GD & GCNII   |      OOM       |        OOM        |       55.28       |     **72.83**     |
> | GAS & GCN    |   95.35±0.01   |    98.91±0.03     |    53.44±0.11     |    71.54±0.19     |
> | GAS & GCNII  |   96.73±0.04   | *99.36*±0.02 |  **55.42**±0.27   |    72.50±0.28     |
> | FM & GCN     |   95.27±0.03   |    98.91±0.01     |    53.48±0.17     |    71.49±0.33     |
> | FM & GCNII   |   96.52±0.06   |    99.34±0.03     |    54.68±0.27     |    72.54±0.27     |
> | LMC & GCN    |   95.44±0.02   |    98.87±0.04     |    53.80±0.14     |    71.44±0.23     |
> | LMC & GCNII  | **96.88**±0.03 |    99.32±0.01     | *55.36*±0.49 | *72.76*±0.22 |
>
>
>
> Besides, our experiments demonstrate that keeping more information can **get the algorithm more robust to *batch* sizes**. **Retrieving the messages recovers the structure information** in the graph such that LMC can resemble the performance of **full-batch gradient descent**---**which does not discard messages during training**---under different batch sizes. Under small batch sizes, as many existing subgraph-wise sampling methods **discard too many messages**, they cannot resemble the full-batch performance. To verify it, we compute the average ratio $|\mathcal{N}(\mathcal{V}\_{\mathcal{B}}) \backslash \mathcal{V}\_{\mathcal{B}}| / |\mathcal{V}\_{\mathcal{B}}|$ to evaluate the correlation between the number of discarded messages and the batch sizes $|\mathcal{V}\_{\mathcal{B}}|$ (the size of the sampled graph) as follows.
>
>
> | Batch Size |  1   |  2   |  5   |  10  |  20  |  40  |
> |:----------:|:----:|:----:|:----:|:----:|:----:|:----:|
> |   Ratio    | 2.09 | 1.95 | 1.71 | 1.40 | 0.95 | 0.46 |
>
> **3. Why does GAS perform better than LMC on some datasets?**
>
> The performance of GAS is marginally better than LMC on PPI and FLICKR. Specifically, the improvements in terms of accuracy are 0.04% and 0.06% on PPI and FLICKR, respectively. The reasons are as follows.
>
> - **FLICKR.** The prediction performance on the FLICKR dataset is unstable, and the corresponding standard deviations are larger than that on other datasets. Due to the random noise in the dataset, the performance of GAS is marginally better than that of LMC.
> - **PPI.** As the feature information is more crucial than the structure information on the PPI dataset, the discarded messages by GAS, CLUSTER-GCN, and GRAPHSAINT do not hurt the prediction performance. On the PPI dataset, both GCN and GCNII additionally introduce the residual connection---which encourages node embeddings to capture the information of themselves rather than their neighbors---to improve the performance.
>
>
> [1] Fey, Matthias, et al. "Gnnautoscale: Scalable and expressive graph neural networks via historical embeddings." International Conference on Machine Learning. PMLR, 2021.

---

> ### Author Response · Authors · 2022-11-18
> **We are looking forward to your further comments.**
>
> Dear Reviewer 7N9L,
>
> Thank you again for your careful reading and insightful comments, which are of great significance for improving our work. The deadline for the discussion stage 1 is approaching, and we are looking forward to your feedback and/or questions. We sincerely hope that our rebuttal has properly addressed your concerns. If so, we would deeply appreciate it if you could raise your score. If not, please let us know your further concerns, and we will continue actively responding to your comments and improving our submission.
>
> Best,
>
> Authors

---

### Official Review · Reviewer_p4iX · 2022-10-24

**Confidence:** 3
**Correctness:** 4
**Technical Novelty And Significance:** 3
**Empirical Novelty And Significance:** 3
**Recommendation:** 8

**Clarity, Quality, Novelty And Reproducibility:**

The paper is easy to understand with clear definition and intuitive illustrations. The novelty comes mostly from the historic compensation part but the authors shows the necessity in ablation studies. Results should be easily reproduced once they publish the code.

**Strength And Weaknesses:**

Pros:
- The paper is well-written with wide coverage of related works.
- The deviate of backward SGD is intuitive and make it easy to follow the further approximation and compensation.
- The claimed theorems and illustration in Figure 1 make the difference between LMC and other methods straightforward.
- Space and time complexity analysis in Appendix adds more concrete supports to the efficiency of LMC
- The uniform sampling and variance reduction technique using neighbors’ historic inform resemble some of the accelerated SGD method, and the authors provided theoretical rate of convergence in a similar way which is good to compare the ideal performance.
- Complete and detailed experiments showing the convergence and efficiency of proposed method.

Cons:
- In equation (7), on the left hand side shouldn’t the gradient for \theta^l depends on V_j^{l + 1} instead of V_j^l  (same for u_{\theta}) ?
- Does the classification performance of GAS and LMC always stay at the same level? What if you change a different base GNN (maybe less expressive and prone to error by gradient variance)?
 - Please add indicator (such as different font, color, etc) in the tables so that its easier to find the winner.
- Is the method universally adapted? What is the performance when applying to small/medium size graph?

**Summary Of The Paper:**

In this work, the authors proposed a subgraph sampling training technique to solve the neighbor-exploding issues commonly seen in GNN training. The proposed methods named Local Message Compensation is claimed to debias the mini-batch gradient in a one-shot sampling setting hence accelerate convergence of training. Both theoretical and experimental results shows the efficacy of the proposed method.


**Summary Of The Review:**

This work is a step forward from FAS. Though the two shares some similarity, the authors shown the improvements with both theoretical guarantee and detailed experiments. Overall I would recommend the paper to be accept.

---

> ### Author Response · Authors · 2022-11-13
> **Response to Reviewer p4iX**
>
> We thank the reviewer for the positive and insightful comments. We respond to each comment as follows and sincerely hope that our rebuttal could properly address your concerns. If so, we would deeply appreciate it if you could raise your score. If not, please let us know your further concerns, and we will continue actively responding to your comments and improving our submission.
>
> **1. About Equation (7).**
>
> The mini-batch gradient $\mathbf{g}\_{\theta^l}(\mathcal{V}\_{\mathcal{B}})$ depends on $\mathbf{V}^l\_{\mathcal{V}\_{\mathcal{B}}}$ and $\left(\nabla\_{\theta^l} u\_{\theta^l}(\mathbf{h}\_j^{l-1}, \mathbf{m}^{l-1}\_{\mathcal{N}(v\_j)}, \mathbf{x}\_j)\right)_{v\_j\in\mathcal{V}\_{\mathcal{B}}}$, as $\mathbf{g}\_{\theta^l}(\mathcal{V}\_{\mathcal{B}})$ is an approximation to the full-batch gradient $\nabla\_{\theta^{l}} \mathcal{L} = \nabla\_{\theta^l}\vec{\mathbf{H}}^l \cdot \nabla\_{\vec{\mathbf{H}}^l}\mathcal{L} = \nabla\_{\theta^l} \vec{f}\_{\theta^l}(\mathbf{H}^{l-1};\mathbf{X}) \cdot \vec{\mathbf{V}}^l$ (see the definition $\mathbf{V}^l \triangleq \nabla\_{\mathbf{H}^l} \mathcal{L}$ and Equation (1)).
>
> Backward SGD uses $(\nabla\_{\theta^l} f\_{\theta^l}(\mathbf{H}^{l-1};\mathbf{X}))\_{\mathcal{V}\_{\mathcal{B}}}=\left(\nabla\_{\theta^l} u\_{\theta^l}(\mathbf{h}\_j^{l-1}, \mathbf{m}^{l-1}\_{\mathcal{N}(v\_j)}, \mathbf{x}\_j)\right)\_{v\_j\in\mathcal{V}\_{\mathcal{B}}}$ and $\mathbf{V}^l_{\mathcal{V}\_{\mathcal{B}}}$ to estimate $\nabla\_{\theta^{l}} \mathcal{L}$.
>
>
>
> **2. Does the classification performance of GAS and LMC always stay at the same level?**
>
>
> Table 3 shows that LMC outperforms GAS by a large margin under small batch sizes (batch size = 1 or 2), where GNNs are prone to error by large gradient variance. Specifically, the improvements in terms of accuracy are 1.09% and 0.77% for GCN and GCNII when batch size = 1, respectively. For your convenience, we quote Table 3 as follows.
>
>
> | Batch size | GAS & GCN | LMC & GCN | GAS & GCNII | LMC & GCNII |
> |:----------:|:---------:|:---------:|:-----------:|:-----------:|
> |     1      |   70.56   | **71.65** |    71.34    |  **72.11**  |
> |     2      |   71.11   |   **71.89**  |    72.25    |  **72.55**  |
> |     5      | **71.99** |   71.84   |    72.23    |  **72.87**  |
> |     10     |   71.60   | **72.14** |  **72.82**  |    72.80    |
>
>
>
> **3. What if you change a different base GNN (maybe less expressive and prone to error by gradient variance)?**
>
>
> In the experiments, we use GCN---which is less expressive than many commonly-used GNNs (e.g., MPNN, GIN, and GAT)---as the base GNN. If you know some GNN that is less expressive than GCN, please suggest it to us. We will conduct experiments with it.
>
> Although we can not conduct experiments with the GNN that is less expressive than GCN at the moment, **we believe that LMC still outperforms GAS due to the advantage of convergence**. In terms of expressiveness, **LMC and GAS share the same expressiveness for any GNN** as they use the same forward pass (setting $\beta\_i=0$ in LMC during evaluation).
>
>
>
>
> **4. Please add indicator (such as different font, color, etc) in the tables.**
>
> Thanks for the suggestions. We have added indicators in the tables in the revised paper accordingly.
>
> **5. Is the method universally adapted? What is the performance when applying to small/medium size graph?**
>
> We have added the convergence curves of GD, LMC, and GAS for GCN on three small datasets, i.e., Cora, Citeseer, and PubMed from Planetoid [1]---whose total number of nodes is less than twenty thousand---in Figure 5 in Appendix E.1 in the revised version. **LMC is faster than GAS, especially with a speed-up of 1.5x on the CiteSeer and PubMed datasets.** Notably, the key bottleneck of training efficiency on small datasets is due to the graph sampling rather than forward and backward passes. Thus, GD is faster than GAS and LMC, as it avoids graph sampling by directly using the whole graph.
>
>
>
> [1] Yang, Zhilin, William Cohen, and Ruslan Salakhudinov. "Revisiting semi-supervised learning with graph embeddings." International conference on machine learning. PMLR, 2016.

---

> ### Author Response · Authors · 2022-11-18
> **We are looking forward to your further comments.**
>
> Dear Reviewer p4iX,
>
> Thank you again for your careful reading and insightful comments, which are of great significance for improving our work. The deadline for the discussion stage 1 is approaching, and we are looking forward to your feedback and/or questions. We sincerely hope that our rebuttal has properly addressed your concerns. If so, we would deeply appreciate it if you could raise your score. If not, please let us know your further concerns, and we will continue actively responding to your comments and improving our submission.
>
> Best,
>
> Authors

---

### Official Review · Reviewer_LGPo · 2022-10-31

**Confidence:** 4
**Clarity, Quality, Novelty And Reproducibility:** The paper is easy to understand and s…
**Correctness:** 3
**Technical Novelty And Significance:** 3
**Empirical Novelty And Significance:** 3
**Recommendation:** 6

**Strength And Weaknesses:**

Strength:

1: A reasonable method for retrieval information outside the mini-batches.
2: Providing convergence analysis.
2: Comprehensive experiments show fast convergence.

Weakness:

1) One thing I am confused is that do you need to maintain the all the node embeddings for all the layers? If that is the case, what if the number of layer is too large, would it be OOM issue? Traditional subgraph sampling methods such as ClusterGCN does not need to main such embeddings for all the layers, as it is computed on-the-fly, and thus more memory efficient.

2) how is the subgraph sampling method playing a role during the convergence? For example, what if using random partition vs metis?

3) how is the per epoch time comparing with different methods? The result presented in the paper is mostly on total training time  (or total epochs ) vs accuracy or loss. It would interesting to see per epoch time as well. As I feel seems there are more variables needing to be updated from traditional subgraph sampling method, LMC will need more time per epoch.

4) how the proposed method compared with GraphFM, which seems quite similar to the proposed method.

5) also some ablation over the combination coefficients beta_i.



**Summary Of The Paper:**

This paper proposed a new subgraph sampling method: Local Message Compensation (LMC) for speeding up GNNs training. The general subgraph sampling method will dscard information outside the mini-batches, and thus brings in performance degradation and slow convergence. LMC deals with this issue by retrieving the discarded information in backward passes based on a message passing formulation of backward passes. This paper also provides convergence analysis. The experiment is promising and achieves good convergence speed and accuracy.

**Summary Of The Review:**

The paper is in easy to understand and the proposed method is making sense and achieves good convergence. I am mostly concerning about the memory usage and space to maintain all the layers historical information.

---

> ### Author Response · Authors · 2022-11-13
> **Response to Reviewer LGPo (1/3)**
>
> We thank the reviewer for the positive and insightful comments. We respond to each comment as follows and sincerely hope that our rebuttal could properly address your concerns. If so, we would deeply appreciate it if you could raise your score. If not, please let us know your further concerns, and we will continue actively responding to your comments and improving our submission.
>
> **1. Does LMC maintain all the node embeddings for all the layers? ClusterGCN does not need to maintain such embeddings for all the layers and it is thus more memory efficient.**
>
> The answer is no. In each training iteration, LMC maintains the embeddings of nodes in the sampled mini-batch and their $1$-hop neighbors (denoted by $\mathcal{N}(\mathcal{V}\_{\mathcal{B}})$) for all the layers. As shown in Table 1 in [1] and Table 5, the memory complexity of LMC and ClusterGCN are $\mathcal{O}(n\_{\rm max}L|\mathcal{V}\_{\mathcal{B}}|d)$ and $\mathcal{O}(L|\mathcal{V}\_{\mathcal{B}}|d)$, respectively, which are both linear with the number of layers $L$. For your convenience, we quote Table 5 as follows.
>
> | Method              | Time                                                         | Memory                                                    |
> | ------------------- | ------------------------------------------------------------ | --------------------------------------------------------- |
> | GD and backward SGD | $\mathcal{O}(L(\|\mathcal{E}\|d+\|\mathcal{V}\|d^2))$        | $\mathcal{O}(L\|\mathcal{V}\|d)$                          |
> | CLUSTER             | $\mathcal{O}(L(n\_{\rm max}\|\mathcal{V}\_{\mathcal{B}}\|d+\|\mathcal{V}\_{\mathcal{B}}\|d^2))$ | $\mathcal{O}(L\|\mathcal{V}\_{\mathcal{B}}\|d)$            |
> | GAS                 | $\mathcal{O}(L(n\_{\rm max}\|\mathcal{V}\_{\mathcal{B}}\|d+\|\mathcal{V}\_{\mathcal{B}}\|d^2))$ | $\mathcal{O}(n\_{\rm max}L\|\mathcal{V}\_{\mathcal{B}}\|d)$ |
> | LMC                 | $\mathcal{O}(L(n_{\rm max}\|\mathcal{V}\_{\mathcal{B}}\|d+\|\mathcal{V}\_{\mathcal{B}}\|d^2))$ | $\mathcal{O}(n\_{\rm max}L\|\mathcal{V}\_{\mathcal{B}}\|d)$ |
>
> - As shown in the third-to-last paragraph of Section 5, LMC stores the embeddings of nodes outside $\mathcal{N}(\mathcal{V}\_{\mathcal{B}})$ in RAM or hard drive storage rather than GPU memory. Therefore, LMC maintains a small number of active node embeddings rather than all node embeddings in forward and backward passes.
> - ClusterGCN maintains embeddings for all the layers in backward passes, which is implemented by autograd packages. For example, for the $l$-th layer $\mathbf{Z}^{l}=\mathbf{A}\mathbf{H}^{l-1}\mathbf{W}^{l}$ in GCN, if we know the gradient $\nabla\_{\mathbf{Z}^{l}} \mathcal{L}$, the autograd packages such as PyTorch compute the gradient $\nabla\_{\mathbf{W}^{l}} \mathcal{L}= (\mathbf{A}\mathbf{H}^{l-1})^{\top} \nabla\_{\mathbf{Z}^{l}} \mathcal{L}$ by the chain rule. Therefore, ClusterGCN maintains the aggregation embeddings $\mathbf{A}\mathbf{H}^{l-1}$ for all $l=1,2,\dots,L$ in backward passes.

---

> > ### Author Response · Authors · 2022-11-13
> > **Response to Reviewer LGPo (2/3)**
> >
> > **2. How is subgraph sampling (e.g., sampling under random or METIS partition) playing a role during the convergence?**
> >
> > The subgraph sampling strategies affect the biases of mini-batch gradients in LMC and hence the convergence. Specifically, sampling under METIS partition increases the variance and decreases the bias of the mini-batch gradients in LMC. Therefore, the subgraph sampling strategy of LMC, ClusterGCN [1], and GAS [2] is a trade-off between the two strategies, i.e., partitioning many clusters by METIS and uniformly sampling them to construct a subgraph.
> >
> > - **Sampling under METIS partition increases the variance of mini-batch gradients**. Sampling under METIS partition encourages the connections between the sampled nodes, whose properties, including auxiliary variables in backward passes, are similar. Therefore, sampling under METIS partition is a cluster sampling strategy, which suffers from a larger variance of gradients than uniform sampling (i.e., sampling under random partition). For example, if we aim to compute the mean of a set $\mathcal{S}=\{1,1,1,1,2,2,2,2,3,3,3,3\}$ by the mean of a sampled batch of elements, the cluster sampling strategy first partition $\mathcal{S}$ into three sets $\mathcal{S}\_1=\{1,1,1,1\}$, $\mathcal{S}\_2=\{2,2,2,2\}$, and $\mathcal{S}\_3=\{3,3,3,3\}$. Then, the cluster sampling strategy outputs $1={\rm mean}(\mathcal{S}\_1)$, $2={\rm mean}(\mathcal{S}\_2)$, and $3={\rm mean}(\mathcal{S}\_3)$ uniformly. If the variance of the cluster sampling strategy is $\sigma^2$, then the variance of uniformly sampling directly from $\mathcal{S}$ is about $\sigma^2/4$. To verify this claim, we report the relative errors of backward SGD (an unbiased baseline introduced in Section 4.2) for a GCN with three layers on the Ogbn-arxiv dataset under difference partitions in the following table.
> >
> >     |                  | Errors at Layer 1 | Errors at Layer 2 | Errors at Layer 3 |
> >     |:---------------- |:-----------------:|:-----------------:|:-----------------:|
> >     | random partition |       0.021       |       0.022       |       0.023       |
> >     | METIS            |       0.304       |       0.334       |       0.343       |
> > - **Sampling under METIS partition decreases the biases of mini-batch gradients**. As shown in [2], sampling under METIS partition minimizes the inter-connectivity between mini-batches to alleviate the errors of the message compensations $\mathbf{C}\_f$ and $\mathbf{C}\_b$, which introduce the bias of the mini-batch gradients.
> >
> >
> > **3. How is the per epoch time comparing with different methods?**
> >
> > We report the per epoch time in Table 6 in Appendix E.2 in the revised version. As shown in Table 6, **the training time per epoch of LMC is comparable with GAS.** For your convenience, we quote Table 6 as follows.
> >
> > | Dataset & GNN      | CLUSTER |   GAS    |  FM  |   LMC    |
> > |:------------------ |:-------:|:--------:|:----:|:--------:|
> > | Ogbn-arxiv & GCN   |  0.51   |   0.46   | 0.75 | **0.45** |
> > | FLICKR & GCN       |  0.33   |   0.29   | 0.45 | **0.26** |
> > | REDDIT & GCN       |  2.16   | **2.11** | 5.67 |   2.28   |
> > | PPI & GCN          |  0.84   | **0.61** | 0.78 |   0.62   |
> > | Ogbn-arxiv & GCNII |   ---   |   0.92   | 1.02 | **0.91** |
> > | FLICKR & GCNII     |   ---   | **1.32** | 1.44 |   1.33   |
> > | REDDIT & GCNII     |   ---   | **4.47** | 7.89 |   4.71   |
> > | PPI & GCNII        |   ---   | **4.61** | 5.38 |   4.77   |
> >
> >
> > Compared with GAS, LMC additionally accesses historical auxiliary variables. Inspired by GAS [2], we use the concurrent mini-batch execution to access historical auxiliary variables asynchronously. Moreover, from the convergence analysis of LMC, we can sample clusters to construct fixed subgraphs at the preprocessing step (Line 2 in Algorithm 1) rather than sample clusters to construct various subgraphs at each training step. This further avoids sampling costs.

---

> > > ### Author Response · Authors · 2022-11-13
> > > **Response to Reviewer LGPo (3/3)**
> > >
> > > **4. The comparison of LMC with GraphFM.**
> > >
> > > We have added GraphFM in Figure 2, and Tables 1 and 2 in the revised version. As shown in Tables 1 and 2, **LMC outperforms GraphFM in terms of prediction accuracy with higher efficiency and comparable memory consumption.** For your convenience, we quote the parts related to LMC and GraphFM of Tables 1 and 2 as follows.
> > >
> > > **Table 1:** Prediction performance of GraphFM and LMC on large graph datasets.
> > > | Method & GNN |   REDDIT   |    PPI     |   FLICKR   | Ogbn-arxiv |
> > > |:------------ |:----------:|:----------:|:----------:|:----------:|
> > > | FM & GCN     | 95.27±0.03 | 98.91±0.01 | 53.48±0.17 | 71.49±0.33 |
> > > | FM & GCNII   | 96.52±0.06 | 99.34±0.03 | 54.68±0.27 | 72.54±0.27 |
> > > | LMC & GCN    | 95.44±0.02 | 98.87±0.04 | 53.80±0.14 | 71.44±0.23 |
> > > | LMC & GCNII  | **96.88**±0.03 | 99.32±0.01 | *55.36*±0.49 | *72.76*±0.22 |
> > >
> > > **Table 2:** Efficiency of GraphFM and LMC.
> > > |                    | Epochs    |           | Runtime |         | Memory |      |
> > > | ------------------ | --------- | --------- | ------- | ------- | ------ | ---- |
> > > | **Dataset & GNN**  | FM        | LMC       | FM      | LMC     | FM     | LMC  |
> > > | Ogbn-arxiv & GCN   | 152.4     | **124.4** | 115     | **55**  | 460    | 557  |
> > > | FLICKR & GCN       | 400.0     | **334.2** | 181     | **85**  | 380    | 376  |
> > > | REDDIT & GCN       | 400.0     | **166.8** | 2269    | **381** | 1644   | 1829 |
> > > | PPI & GCN          | **286.4** | 290.2     | 224     | 179     | 218    | 267  |
> > > | Ogbn-arxiv & GCNII | 373.6     | **197.4** | 381     | **178** | 454    | 568  |
> > > | FLICKR & GCNII     | 400.0     | 356       | 576     | 475     | 402    | 468  |
> > >
> > >
> > > **5. The ablation over the combination coefficients $\beta\_i$.**
> > >
> > > We have added the ablation over the combination coefficients $\beta\_i = score(i)\alpha$ in Tables 8 and 9 in Appendix E.4 in the revised version. For your convenience, we quote Tables 8 and 9 as follows.
> > >
> > >
> > > **Table 8:** Prediction performance under different $\alpha$ on the Ogbn-arxiv dataset.
> > > | Batch size | learning rate |  0.0  |  0.2  |    0.4    |  0.6  |  0.8  |    1.0    |
> > > |:----------:|:-------------:|:-----:|:-----:|:---------:|:-----:|:-----:|:---------:|
> > > |     1      |     1e-4      | 71.34 | 71.39 | **71.65** | 71.31 | 70.86 |   70.57   |
> > > |     40     |     1e-2      | 69.85 | 69.12 |   69.89   | 69.61 | 69.82 | **71.44** |
> > >
> > > **Table 9:** Prediction performance under different $score$ on the Ogbn-arxiv dataset.
> > > | Batch size | learning rate | $f(x)=2x-x^2$ | $f(x)=1$ | $f(x)=x^2$ | $f(x)=x$ | $f(x)=\sin(x)$ |
> > > |:----------:|:-------------:|:-------------:|:--------:|:----------:|:--------:|:--------------:|
> > > |     1      |     1e-4      |     **71.35**     |  70.84   | 71.32  |  71.30   |     71.13      |
> > > |     40     |     1e-2      |     67.59     |  **71.44**   |   69.91    |  70.03   |     70.32      |
> > >
> > >
> > > - Under large batch sizes (batch size = 40), LMC achieves the best performance with large $\beta\_i=1$, as large batch sizes improve the quality of the incomplete up-to-date messages.
> > > - Under small batch sizes (batch size = 1), LMC achieves the best performance with small $\beta\_i=0.4 score\_{2x-x^2}(i)$, as small learning rates alleviate the staleness of the historical values.
> > >
> > >
> > >
> > > [1] Chiang, Wei-Lin, et al. "Cluster-gcn: An efficient algorithm for training deep and large graph convolutional networks." Proceedings of the 25th ACM SIGKDD international conference on knowledge discovery & data mining. 2019.
> > >
> > > [2] Fey, Matthias, et al. "Gnnautoscale: Scalable and expressive graph neural networks via historical embeddings." International Conference on Machine Learning. PMLR, 2021.

---

> ### Author Response · Authors · 2022-11-18
> **We are looking forward to your further comments.**
>
> Dear Reviewer LGPo,
>
> Thank you again for your careful reading and insightful comments, which are of great significance for improving our work. The deadline for the discussion stage 1 is approaching, and we are looking forward to your feedback and/or questions. We sincerely hope that our rebuttal has properly addressed your concerns. If so, we would deeply appreciate it if you could raise your score. If not, please let us know your further concerns, and we will continue actively responding to your comments and improving our submission.
>
> Best,
>
> Authors

---

### Official Review · Reviewer_DdvT · 2022-11-01

**Confidence:** 3
**Correctness:** 3
**Technical Novelty And Significance:** 3
**Empirical Novelty And Significance:** 3
**Recommendation:** 8

**Clarity, Quality, Novelty And Reproducibility:**

This manuscript is excellently written. Their method is clearly explained in the notations, equations, and descriptions. Their novelties include a new backward SGD technique and theoretical guarantees.

**Strength And Weaknesses:**

Strength:
(1) Their approach helps to avoid the neighbor explosion problem to a certain extent.
(2) The convergence of their method is provable.

Weaknesses:
1. one typo is that in table 1 GraphSAINT's 97.0 reaches the best.
2. It seems to me that the authors claim that their method is more efficient than other methods, which means the "proportion of reserved messages (%) in forward and backward passes" based on their response, but in terms of memory in Table 2, for example, 1892 for LMC is about 2 times more than 1193 for CLUSTER and 557 is about 1.3 times more than 424, which may confuse others in terms of memory efficiency. It would be clearer to modify Table 2 and add "proportion of reserved messages (%) in forward and backward passes" to it.

**Summary Of The Paper:**

The authors proposed a novel subgraph-wise sampling method, Local Message Compensation(LMC), to accelerate the training of GNNs on large-scale graphs. They mainly focus on solving the neighbor explosion problem. Their main concern is finding a solution to the neighbor explosion issue. I think their research has a significant impact on the advancement of GNNs.

**Summary Of The Review:**

Even though their work aids in the development of GNNs, these are insufficient for being strongly accepted.

---

> ### Author Response · Authors · 2022-11-13
> **Response to Reviewer DdvT (1/2)**
>
> We thank the reviewer for the insightful and valuable comments. We respond to your comments as follows and sincerely hope that our rebuttal could properly address your concerns. If so, we would deeply appreciate it if you could raise your score（"3: reject, not good enough"). If not, please let us know your further concerns, and we will continue actively responding to your comments and improving our submission.
>
>
> **1. How is LMC more efficient than the other methods even though it uses more memory? From the experiments, LMC does not improve much in terms of accuracy and time.**
>
>
> The additional memory usage of LMC is marginal due to the reasons as follows.
>
> - **LMC is more memory-efficient and faster than the full-batch GD**---which is provably convergent by keeping all the available neighborhood information in both forward and backward passes---especially with **a memory consumption reduction of 79%** under batch size = 1 on the REDDIT dataset.
>
>   - **The memory consumption of LMC is significantly lower than GD** (especially with **a memory consumption reduction of 79%** under batch size = 1 on REDDIT dataset), although both LMC and GD use all the available neighborhood information in both forward and backward passes. Following Table 3 in [1], we report the GPU memory consumption of different methods in Table 7 in Appendix E.3 in the revised version. For your convenience, we quote it as follows.
>
>     **Table 7:** GPU memory consumption (MB) and the proportion of reserved messages (\%) in forward and backward passes of GD, CLUSTER, GAS, and LMC for training GCN. *Default* indicates the default batch size used in the codes and toolkits of GAS.
>
>     | Batch size | Methods |      Ogbn-arxiv       |        FLICKR         |         REDDIT         |          PPI          |
>     | :--------: | :-----: | :-------------------: | :-------------------: | :--------------------: | :-------------------: |
>     | full-batch |   GD    | 681/**100**%/**100**% | 411/**100**%/**100**% | 2067/**100**%/**100**% | 605/**100**%/**100**% |
>     |     1      | CLUSTER |    **177**/67%/67%    |    **138**/57%/57%    |    **428**/35%/35%     |    **189**/90%/90%    |
>     |     1      |   GAS   |   178/**100**%/67%    |   168/**100**%/57%    |    482/**100**%/35%    |   190/**100**%/90%    |
>     |     1      |   LMC   | 207/**100**%/**100**% | 177/**100**%/**100**% | 610/**100**%/**100**%  | 197/**100**%/**100**% |
>     |  Default   | CLUSTER |      424/83%/83%      |      310/77%/77%      |      1193/65%/65%      |      212/91%/91%      |
>     |  Default   |   GAS   |   452/**100**%/83%    |   375/**100**%/77%    |   1508/**100**%/65%    |   214/**100**%/91%    |
>     |  Default   |   LMC   | 557/**100**%/**100**% | 376/**100**%/**100**% | 1829/**100**%/**100**% | 267/**100**%/**100**% |
>
>     As shown in Table 7, LMC makes full use of all sampled nodes in both forward and backward passes, which is the same as GD. However, existing subgraph-wise sampling methods **discard messages to save memory**, which **poses significant challenges to their convergence behaviors**, e.g., convergence speeds and the robustness to batch sizes.
>
>
>
>   - **LMC is scalable but GD is not**. As shown in Table 5 in Appendix B, the memory complexity of LMC is $\mathcal{O}(n\_{\rm max}L|\mathcal{V}\_{\mathcal{B}}|d)$, which is linear with the size of mini-batches $|\mathcal{V}\_{\mathcal{B}}|$ rather than that of the whole graph. However, the memory complexity of GD is $\mathcal{O}(L|\mathcal{V}|d)$, which is linear with the size of the whole graph $|\mathcal{V}|$. For your convenience, we quote Table 5 as follows.
>
>       | Method              | Time                                                         | Memory                                                    |
>       | ------------------- | ------------------------------------------------------------ | --------------------------------------------------------- |
>       | GD and backward SGD | $\mathcal{O}(L(\|\mathcal{E}\|d+\|\mathcal{V}\|d^2))$        | $\mathcal{O}(L\|\mathcal{V}\|d)$                          |
>       | CLUSTER             | $\mathcal{O}(L(n\_{\rm max}\|\mathcal{V}\_{\mathcal{B}}\|d+\|\mathcal{V}_{\mathcal{B}}\|d^2))$ | $\mathcal{O}(L\|\mathcal{V}\_{\mathcal{B}}\|d)$            |
>       | GAS                 | $\mathcal{O}(L(n\_{\rm max}\|\mathcal{V}_{\mathcal{B}}\|d+\|\mathcal{V}\_{\mathcal{B}}\|d^2))$ | $\mathcal{O}(n\_{\rm max}L\|\mathcal{V}\_{\mathcal{B}}\|d)$ |
>       | LMC                 | $\mathcal{O}(L(n\_{\rm max}\|\mathcal{V}\_{\mathcal{B}}\|d+\|\mathcal{V}\_{\mathcal{B}}\|d^2))$ | $\mathcal{O}(\_{\rm max}L\|\mathcal{V}\_{\mathcal{B}}\|d)$ |
>
>
>   - **LMC is significantly faster than GD**, as shown in Figure 2.

---

> > ### Author Response · Authors · 2022-11-13
> > **Response to Reviewer DdvT (2/2)**
> >
> > - **LMC significantly outperforms GAS by a large margin in terms of accuracy** under small batch sizes, as shown in Table 3. Specifically, the improvements in terms of accuracy are 1.09% and 0.77% for GCN and GCNII when batch size = 1, respectively. For your convenience, we quote Table 3 as follows.
> >
> > | Batch size | GAS & GCN | LMC & GCN | GAS & GCNII | LMC & GCNII |
> > |:----------:|:---------:|:---------:|:-----------:|:-----------:|
> > |     1      |   70.56   | **71.65** |    71.34    |  **72.11**  |
> > |     2      |   71.11   |   **71.89**  |    72.25    |  **72.55**  |
> >
> >
> > - **LMC is significantly faster than existing subgraph-wise sampling methods (e.g., GAS)**, especially with a **speed-up of 2x** on the REDDIT dataset, as shown in Table 2. For your convenience, we quote the **Runtime** columns of Table 2 and compute the speed-ups as follows.
> >
> >     | **Dataset & GNN**  | GAS     |  LMC    | Speed-ups  |
> >     |:------------------ |:-------:|:-------:|:-------:|
> >     | Ogbn-arxiv & GCN   | 79      | **55**  |1.44|
> >     | FLICKR & GCN       | 117     | **85**  |1.38|
> >     | REDDIT & GCN       | 790     | **381** |2.07|
> >     | PPI & GCN          | **179** | **179** |1.00|
> >     | Ogbn-arxiv & GCNII |  218    | **178** |1.22|
> >     | FLICKR & GCNII     | **465** |  475    |0.98|
> >
> > -  **LMC and GAS share the same memory complexity**, i.e., $\mathcal{O}(n\_{\rm max}L|\mathcal{V}\_{\mathcal{B}}|d)$, as shown in Table 5 in Appendix B. For your convenience, we quote Table 5 as follows.
> >
> >       | Method              | Time                                                         | Memory                                                    |
> >     | ------------------- | ------------------------------------------------------------ | --------------------------------------------------------- |
> >     | GD and backward SGD | $\mathcal{O}(L(\|\mathcal{E}\|d+\|\mathcal{V}\|d^2))$        | $\mathcal{O}(L\|\mathcal{V}\|d)$                          |
> >     | CLUSTER             | $\mathcal{O}(L(n\_{\rm max}\|\mathcal{V}\_{\mathcal{B}}\|d+\|\mathcal{V}\_{\mathcal{B}}\|d^2))$ | $\mathcal{O}(L\|\mathcal{V}\_{\mathcal{B}}\|d)$            |
> >     | GAS                 | $\mathcal{O}(L(n\_{\rm max}\|\mathcal{V}\_{\mathcal{B}}\|d+\|\mathcal{V}\_{\mathcal{B}}\|d^2))$ | $\mathcal{O}(n\_{\rm max}L\|\mathcal{V}\_{\mathcal{B}}\|d)$ |
> >     | LMC                 | $\mathcal{O}(L(n\_{\rm max}\|\mathcal{V}\_{\mathcal{B}}\|d+\|\mathcal{V}\_{\mathcal{B}}\|d^2))$ | $\mathcal{O}(n\_{\rm max}L\|\mathcal{V}\_{\mathcal{B}}\|d)$ |
> >
> >     Compared with the memory usage of GD, their practical memory usages are similar, and the difference is due to the implementation. As **the additional memory does not lead to the out-of-memory issue**, we focus on convergence speeds in practice. We plan to bridge the gap between theory and implementation in the future.
> >
> >
> >
> >
> >
> > [1] Fey, Matthias, et al. "Gnnautoscale: Scalable and expressive graph neural networks via historical embeddings." International Conference on Machine Learning. PMLR, 2021.

---

> ### Author Response · Authors · 2022-11-18
> **We are looking forward to your further comments.**
>
> Dear Reviewer DdvT,
>
> Thank you again for your careful reading and insightful comments, which are of great significance for improving our work. The deadline for the discussion stage 1 is approaching, and we are looking forward to your feedback and/or questions. We sincerely hope that our rebuttal has properly addressed your concerns. If so, we would deeply appreciate it if you could raise your score. If not, please let us know your further concerns, and we will continue actively responding to your comments and improving our submission.
>
> Best,
>
> Authors

---

> > ### Comment · Reviewer_DdvT · 2022-11-19
> > **Response to Authors**
> >
> > Thank you for your detailed reply. After reading your response, I have changed my score.
> >
> > Best,

---

> > > ### Author Response · Authors · 2022-11-19
> > > **Thank you for your kind support**
> > >
> > > Dear Reviewer DdvT,
> > >
> > > Thank you for your kind support and helping us improve the paper! We enjoy communicating with you and appreciate your valuable suggestions!
> > >
> > > Best,
> > >
> > > Authors

---

### Decision · Program_Chairs · 2023-01-20

**Decision:**

Accept: notable-top-25%

**Justification For Why Not Higher Score:**

I think that the paper is interesting but it is not solving a problem that is central enough to be an oral

**Justification For Why Not Lower Score:**

The paper contains both interesting experimental and theoretical insights and it is well-written.

**Metareview: Summary, Strengths And Weaknesses:**

The authors introduce a new subgraph sampling method, Local Message Compensation (LMC), to efficiently train GNN. Interestingly this model directly address one of the main challenges in training GNN the well-known neighbor explosion problem.

The paper is well-written and the main idea behind the result are very well-explained. The main idea behind the proposed technique is to debias the mini-batch gradient in a one-shot sampling setting to accelerate convergence of training. The authors show that their method works well both theoretically than experimentally.

Overall, the paper is a nice contribution to ICLR and I suggest to accept it.

**Note From Pc:**

if the above contains the word "oral" or "spotlight" please see: "oral" presentation means -> notable-top-5% and "spotlight" means -> notable-top-25%. As stated in our emails, we are disassociating presentation type from AC recommendations